# Learning spatiotemporal signals using a recurrent spiking network that discretizes time

**Amadeus Maes**[1], **Mauricio Barahona**[2], **Claudia Clopath**[1] *

**1** Department of Bioengineering, Imperial College London, London, United Kingdom, **2** Department of Mathematics, Imperial College London, London, United Kingdom

* c.clopath@imperial.ac.uk

## Abstract

Learning to produce spatiotemporal sequences is a common task that the brain has to solve. The same neurons may be used to produce different sequential behaviours. The way the brain learns and encodes such tasks remains unknown as current computational models do not typically use realistic biologically-plausible learning. Here, we propose a model where a spiking recurrent network of excitatory and inhibitory spiking neurons drives a read-out layer: the dynamics of the driver recurrent network is trained to encode time which is then mapped through the read-out neurons to encode another dimension, such as space or a phase. Different spatiotemporal patterns can be learned and encoded through the synaptic weights to the read-out neurons that follow common Hebbian learning rules. We demonstrate that the model is able to learn spatiotemporal dynamics on time scales that are behaviourally relevant and we show that the learned sequences are robustly replayed during a regime of spontaneous activity.

## Author summary

The brain has the ability to learn flexible behaviours on a wide range of time scales. Previous studies have successfully built spiking network models that learn a variety of computational tasks, yet often the learning involved is not biologically plausible. Here, we investigate a model that uses biological-plausible neurons and learning rules to learn a specific computational task: the learning of spatiotemporal sequences (i.e., the temporal evolution of an observable such as space, frequency or channel index). The model architecture facilitates the learning by separating the temporal information from the other dimension. The time component is encoded into a recurrent network that exhibits sequential dynamics on a behavioural time scale, and this network is then used as an engine to drive the read-out neurons that encode the spatial information (i.e., the second dimension). We demonstrate that the model can learn complex spatio-temporal spiking dynamics, such as the song of a bird, and replay the song robustly spontaneously.

**Data Availability Statement:** All code is available from the modelDB database at the URL http://modeldb.yale.edu/257609.

**Funding:** This work has been funded by EPSRC (EP/N014529/1) (to MB) and by BBSRC (BB/

N013956/1 and BB/N019008/1), Wellcome Trust (200790/Z/16/Z), the Simons Foundation (564408), and the EPSRC (EP/R035806/1) (to CC). The funders had no role in study design, data collection and analysis, decision to publish, or preparation of the manuscript.

**Competing interests:** The authors have declared that no competing interests exist.

## Introduction

Neuronal networks perform flexible computations on a wide range of time scales. While individual neurons operate on the millisecond time scale, behaviour time scales typically span from a few milliseconds to hundreds of milliseconds and longer. Building functional models that bridge this time gap is of increasing interest [1], especially now that the activity of many neurons can be recorded simultaneously [2, 3]. Many tasks and behaviours in neuroscience consist of learning and producing flexible spatiotemporal sequences, e.g. a 2-dimensional pattern with time on the x-axis and any other observable on the y-axis which we denote here in general terms as the "spatial information". For example, songbirds produce their songs through a specialized circuit: neurons in the HVC nucleus burst sparsely at very precise times to drive the robust nucleus of the arcopallium which in its turn drives motor neurons [4, 5]. For different motor tasks, sequential neuronal activity is recorded in various brain regions [6–10], and while the different tasks involve different sets of muscles, the underlying computation on a more fundamental level might be similar [11].

Theoretical and computational studies have shown that synaptic weights of recurrent networks can be set appropriately so that dynamics on a wide range of time scales is produced [12–14]. In general, these synaptic weights are engineered to generate a range of interesting dynamics. In slow-switching dynamics, for instance, the wide range of time scales is produced by having stochastic transitions between clusters of neurons [15]. Another example is sequential dynamics, where longer time scales are obtained by clusters of neurons that activate each other in a sequence. This sequential dynamics can emerge by a specific connectivity in the excitatory neurons [16, 17] or in the inhibitory neurons [18, 19]. However, it is unclear how the brain learns these dynamics, as most of the current approaches use non biologically plausible ways to set or "train" the synaptic weights. For example, FORCE training [20–22] or backpropagation through time [23] use non-local information either in space or in time to update weights. Such information is not available to the synaptic connection, which only has access to the presynaptic and postsynaptic variables at the current time.

Here, we propose to learn a spatiotemporal task over biologically relevant time scales using a spiking recurrent network driving a read-out layer where the neurons and synaptic plasticity rules are biologically plausible. Specifically, all synapses are plastic under typical spike-timing dependent Hebbian learning rules [12, 24]. Our model architecture decomposes the problem into two parts. First, we train a recurrent network to generate a sequential activity which serves as a temporal backbone so that it operates as a 'neuronal clock' driving the downstream learning. The sequential activity is generated by clusters of neurons activated one after the other: as clusters are highly recurrently connected, each cluster undergoes reverberating activity that lasts longer than neural time scale so that the sequential cluster activation is long enough to be behaviourally relevant. This construction allows us to bridge the neural and the behavioural time scales. Second, we use Hebbian learning to encode the target spatiotemporal dynamics in the read-out neurons. In this way, the recurrent network encodes time and the read-out neurons encode 'space'. As discussed above, we use the term 'space' to denote a temporally-dependent observable, be it spatial position, or phase, or a time-dependent frequency, or a more abstract state-space. Similar to the liquid state-machine, where the activity in a recurrent network is linearly read-out by a set of neurons, we can learn different dynamics in parallel in different read-out populations [25]. We also show that learning in the recurrent network is stable during spontaneous activity and that the model is robust to synaptic failure.

## Results

### Model architecture

The model consists of two separate modules: a recurrent network and a read-out layer (Fig 1A). Learning happens in two stages. In the first stage, we learn the weights of the recurrent network so that the network exhibits a sequential dynamics. The ensuing recurrent neuronal network (RNN) effectively serves as a temporal backbone driving the learning of the downstream read-out layer. In the second stage, a target sequence is learned in the read-out layer.

**Architecture.** The recurrent network is organized in $C$ clusters of excitatory neurons and a central cluster of inhibitory neurons. All excitatory neurons follow adaptive exponential

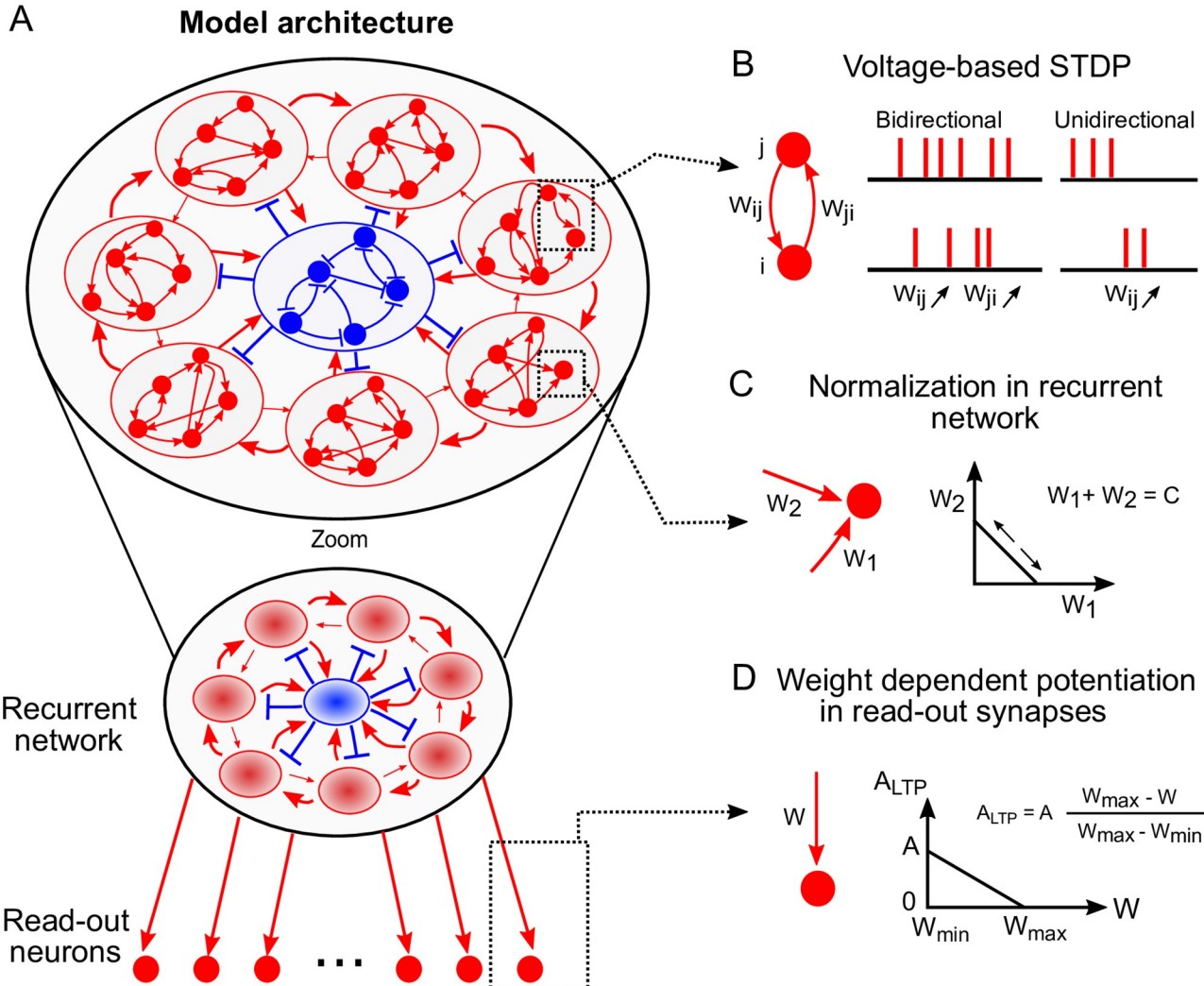

**Fig 1. Model architecture.** (A) The recurrent network consists of both inhibitory (in blue) and excitatory (in red) neurons. The connectivity is sparse in the recurrent network. The temporal backbone is established in the recurrent network after a learning phase. Inset: zoom of recurrent network showing the macroscopic recurrent structure after learning, here for 7 clusters. The excitatory neurons in the recurrent network project all-to-all to the read-out neurons. The read-out neurons are not interconnected. (B) All excitatory to excitatory connections are plastic under the voltage-based STDP rule (see Methods for details). The red lines are spikes of neuron $j$ (top) and neuron $i$ (bottom). When neurons $j$ and $i$ are very active together, they form bidirectional connections strengthening both $W_{ij}$ and $W_{ji}$. Connections $W_{ij}$ are unidirectionally strengthened when neuron $j$ fires before neuron $i$. (C) The incoming excitatory weights are $L1$ normalized in the recurrent network, i.e. the sum of all incoming excitatory weights is kept constant. (D) Potentiation of the plastic read-out synapses is linearly dependent on the weight. This gives weights a soft upper bound.

integrate-and-fire dynamics [26] while all inhibitory neurons follow a leaky integrate-and-fire dynamics. The inhibitory neurons in the RNN prevent pathological dynamics. The aim of this module is to discretize time into $C$ sequential intervals, associated with each of the $C$ clusters. This is achieved by learning the weights of the recurrent network. The neurons in the excitatory clusters then drive read-out neurons through all-to-all feedforward connections. The read-out neurons are not interconnected. The target sequence is learned via the weights between the driver RNN and the read-out neurons.

**Plasticity.** In previous models, the learning schemes are typically not biologically plausible because the plasticity depends on non-local information. Here, however, we use the voltage-based STDP plasticity rule in all the connections between excitatory neurons (Fig 1B). This is paired with weight normalization in the recurrent network (Fig 1C) and weight dependent potentiation in the read-out synapses (Fig 1D). Inhibitory plasticity [27] finds good parameters aiding the sequential dynamics (S5 Fig).

**Learning scheme.** During the first stage of learning, all neurons in each cluster receive the same input in a sequential manner. As a result of this learning stage, the recurrent spiking network displays a sequential dynamics of the $C$ clusters of excitatory neurons. Neurons within each cluster spike over a time interval (while all neurons from other clusters are silent), with the activity switching clusters at points $t = [t_0, t_1, ..., t_C]$ so that cluster $i$ is active during time interval $[t_{i-1}, t_i]$. Thus, time is effectively discretized in the RNN.

During the second stage of learning, the read-out neurons receive input from a set of excitatory supervisor neurons. The discretization of time enables Hebbian plasticity to form strong connections from the neurons in the relevant time bin to the read-out neurons. For instance, if we want to learn a signal which is 'on' during $[t_{i-1}, t_i]$ and 'off' otherwise, a supervisor neuron can activate the read-out neuron during that time interval so that connections from cluster $i$ to the read-out neuron are potentiated through activity (who fires together, wires together). This means that, after learning, the read-out neuron will be activated when cluster $i$ is activated. In general, the read-out layer learns a multivariate signal of time, i.e., the neurons in the read-out layer encode the $D$ different dimensions of the target signal: $t \to \phi(t) = [\phi_1(t), \phi_2(t), ..., \phi_D(t)]$.

## A recurrent network that encodes discrete time

We give here further details of the first learning stage, where a recurrent network is trained to produce a sequential dynamics. To this end, we initialize the weight matrix so that each synaptic weight between two neurons is non-zero with probability $p$. The weights that are zero remain zero at all times, i.e. the topology is fixed. We set the initial values of the non-zero weights in the recurrent network such that the dynamics is irregular and asynchronous (i.e., a balanced network, see Methods for details).

We stimulate the $C$ clusters with an external input in a sequential manner (Fig 2A): neurons in cluster $i$ each receive external Poisson spike trains (rate of 18 k$Hz$ for 10 m$s$, assuming a large input population). After this, there is a time gap where no clusters receive input (5 m$s$). This is followed by a stimulation of cluster $i + 1$. This continues until the last cluster is reached and then it links back to the first cluster (i.e. a circular boundary condition). During the stimulation, neurons in the same cluster fire spikes together strengthening the intra-cluster connections bidirectionally through the voltage-based STDP rule [24, 28]. Additionally, there is a pre/post pairing between adjacent clusters. Neurons in cluster $i + 1$ fire after neurons in cluster $i$. The weights from cluster $i$ to cluster $i + 1$ strengthen unidirectionally (Fig 2B). If the time gap between sequential stimulations is increased during the training phase, so that the gap becomes too long with respect to the STDP time window, then there is no pre/post pairing between clusters and the ensuing dynamics loses its sequential nature and becomes a slow-

## A  Learning a feedforward structure through the recurrent network

**Sequential external input during learning**

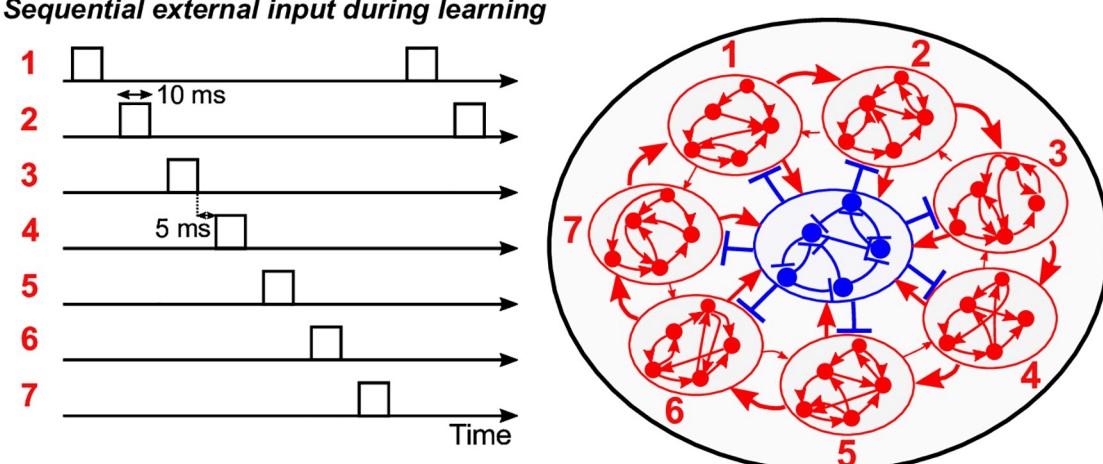

## B  Weight matrix of RNN after learning

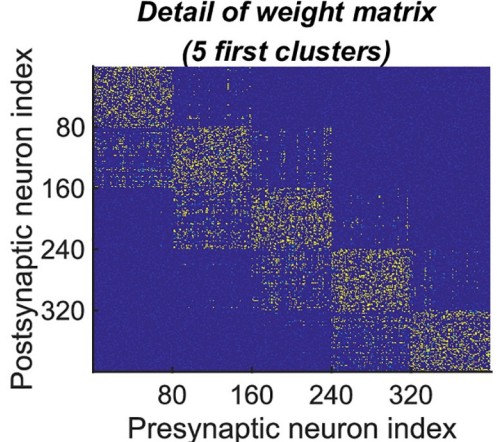

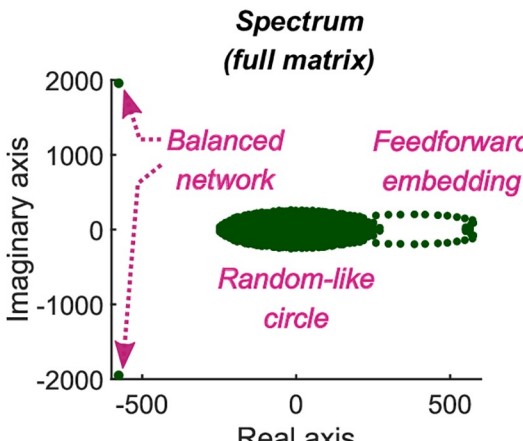

## C  After learning RNN discretizes time under spontaneous random activity

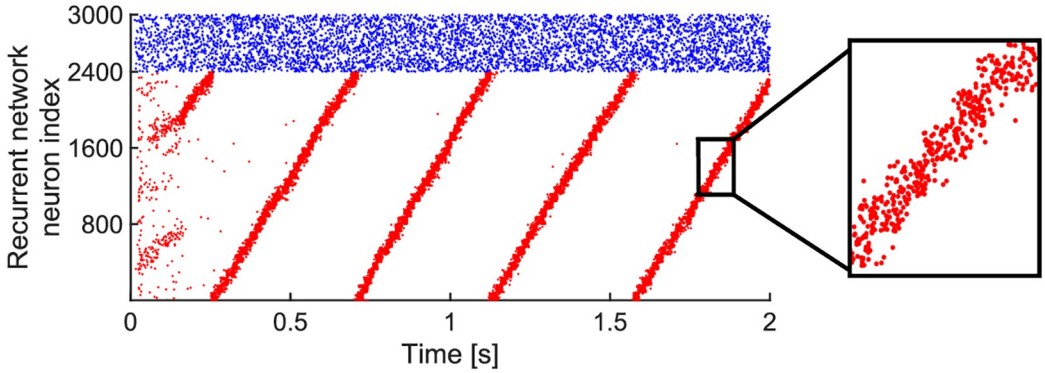

**Fig 2. Learning a sequential dynamics stably under plasticity.** (A) The excitatory neurons receive sequential clustered inputs. Excitatory neurons are grouped in 30 disjoint clusters of 80 neurons each. (7 clusters shown in the cartoon for simplicity) (B) The weight matrix after training (only the first five clusters shown) exhibits the learned connectivity structure, e.g., neurons within cluster 1 are highly interconnected and also project to neurons in cluster 2, same for cluster 2 to cluster 3, etc. The spectrum of the full weight matrix after training shows most eigenvalues in a circle in the complex plane (as in a random graph) with two other eigenvalues signifying the balancing of the network, and a

series of dominant eigenvalues in pairs that encode the feedforward embedding. (C) Raster plot of the total network consisting of 2400 excitatory (in red) and 600 inhibitory (in blue) neurons. After learning, the spontaneous dynamics exhibits a stable periodic trajectory 'going around the clock'. The excitatory clusters discretize time (see zoom) and the network has an overall period of about 450 m$s$.

switching dynamics [12, 15] (S1 Fig). In slow-switching dynamics, clusters of neurons are active over long time scales, but both the length of activation and the switching between clusters is random. This is because the outgoing connections from cluster $i$ to all other clusters are the same in a slow-switching network. To summarize, a connectivity structure emerges through biophysically plausible potentiation by sequentially stimulating the clusters of excitatory neurons in the recurrent network. When the gap between activation intervals is sufficiently small compared to the STDP window, the connectivity structure is such that intra-cluster weights and the weights from successive clusters $i \rightarrow i + 1$ are strong.

After the synaptic weights have converged, the external sequential input is shut-down and spontaneous dynamics is simulated so that external excitatory Poisson spike trains without spatial or temporal structure drive the RNN. Under such random drive, the sequence of clusters reactivates spontaneously and ensures that both the intra-cluster and the connections from cluster $i$ to cluster $i + 1$ remain strong. In general, the interaction between plasticity, connectivity and spontaneous dynamics can degrade the learned connectivity and lead to unstable dynamics [29]. To test the stability of the learned connectivity, we track the changes in the off-diagonal weights (i.e. the connections from cluster $i$ to cluster $i + 1$). After the external sequential input is shut-down, we copy the weight matrix and freeze the weights of this copy. We run the dynamics of the recurrent network using the copied frozen weights and apply plastic changes to the original weight matrix. This means that we effectively decouple the plasticity from the dynamics. Indeed, when the dynamics is sequential, the off-diagonal structure is reinforced. When the off-diagonal structure is removed from the frozen copied weight matrix, the dynamics is not sequential anymore. In this case, the off-diagonal structure degrades. We conclude that the connectivity pattern is therefore stable under spontaneous dynamics (S2 Fig).

We next studied how the spectrum of the recurrent weight matrix is linked to the sequential dynamics. In linear systems, the eigenvalues of the connectivity matrix determine the dynamics of the system. In a nonlinear spiking model, the relationship between connectivity and dynamics is less clear. The connectivity after learning can be seen as a low-dimensional perturbation of a random matrix. Such low-dimensional perturbations create outliers in the spectrum [30] and change the dynamics [31]. Here, we have carried out a similar spectral analysis to that presented in [15] (see Fig 7 in the Methods and S2 Fig in the Supplementary material). The weight matrix has most of its eigenvalues in a circle in the complex plane (Fig 2B) with eigenvalues associated both with the balanced nature of the network, but, importantly, also with the sequential structure (Fig 2B). As the temporal backbone develops through learning (as seen in S2 Fig), it establishes a spectral structure in which the pairs of leading eigenvalues with large real parts have almost constant imaginary parts.

A simplified analysis of a reduced weight matrix (where nodes are associated with groups of neurons) shows that the imaginary parts of the dominant eigenvalues depend linearly on the strength of the weights from cluster $i$ to cluster $i + 1$ (see Methods, Fig 7). Hence for this simplified linearised rate model, this results in an oscillatory dynamics where the imaginary part determines the frequency by which the pattern of activation returns due to the periodic excitation pattern. As shown in [15], these properties of the linear system carry over to the nonlinear spiking model, i.e., the imaginary parts of the eigenvalues with large real parts determine the time scales of the sequential activity (S2 Fig).

Under spontaneous activity, each cluster is active for about 15 ms, due to the recurrent connectivity within the cluster. A large adaptation current counteracts the recurrent reverberating activity to turn off the activity reliably. Therefore, as each cluster is spontaneously active in a sequence, the sequence length reaches behavioural time scales (Fig 2C). In summary, the network exhibits sequential dynamics, serving as a temporal backbone where time is discretized over behavioural time scales.

## Learning a non-Markovian sequence

After the sequential temporal backbone is learnt via the RNN, we can then learn a spatiotemporal sequence via the read-out neurons. To achieve this, during the second stage of training, the read-out neurons receive additional input from supervisor neurons and from interneurons (Fig 3A). The supervisor neurons receive an external Poisson input with rate modulated by the target sequence to be learned (Fig 3B).

As a first example, consider a target sequence composed of states *A*, *B*, *C* activated in the following deterministic order: *ABCBA*. This is a non-Markovian state sequence because the transition from state *B* to the next state (*A* or *C*) requires knowledge about the previous state [32], a non trivial task that requires information to be stored about previous network states, potentially over long time periods. Previous studies have proposed various solutions for this task [32, 33]. However, separating the problem of sequence learning in two stages solves this in a natural way.

The recurrent network trained in the first stage (Fig 2) is used to encode time. The underlying assumption is that a starting signal activates both the first cluster of the recurrent network and the external input to the supervisor neurons, which activate the read-out neurons.

After the training period, the interneurons and supervisor neurons stop firing (Fig 3C) and the target sequence is stored in the read-out weight matrix (Fig 3D). During spontaneous activity, clusters in the RNN reactivate in a sequential manner driving the learned sequence in the read-out neurons. Hence the spike sequence of the read-out neurons is a noisy version of the target signal (Fig 3E). Learning the same target signal several times results in slightly different read-out spike sequences each time (S3 Fig). The firing rates of neurons in the read-out corresponds to the target sequence (Fig 3F). In summary, our results show that the model is able to learn simple but non-trivial spatiotemporal signals that are non-Markovian.

## Learning sequences in parallel

We next wondered how multiple spatiotemporal signals can be learned. We hypothesized that, once the temporal backbone is established, multiple spatiotemporal sequences can easily be learned in parallel. As an example, we learn two sequences: *ABCBA* and *DEDED*. Here, *D* and *E* denote two additional read-out neurons (Fig 4A). We assume that the model observes each sequence alternately for 2 seconds at a time (Fig 4B), although in principle it could also been shown simultaneously. After learning, the target sequences are encoded in the read-out weight matrix (Fig 4C). In a regime of spontaneous dynamics the learned sequences can be replayed (Fig 4D). An external inhibitory current to the read-out neurons can control which sequence is replayed. We conclude that multiple sequences can be learned in parallel. Each separate sequence requires a separate set of read-out neurons. As such, the number of read-out neurons required increases linearly with the number of target sequences.

## Properties of the model: Scaling, robustness and temporal variability

We investigate several scaling properties of the network. We first assess how the sequential dynamics in the RNN depends on the cluster size by increasing the number of excitatory neurons in each cluster ($N_C$), preserving the ratio of excitatory to inhibitory neurons ($N_E/N_I$). To

A    Learning a sequence through the read-out neurons

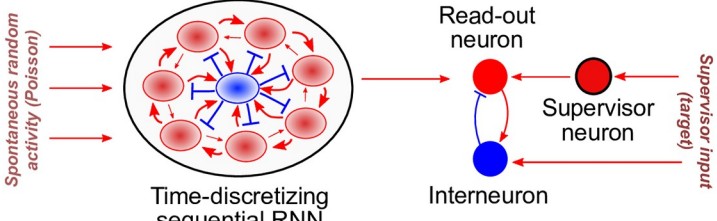

B    Supervisor signal

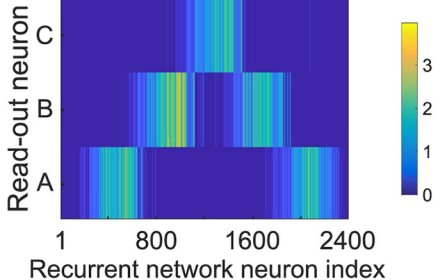

C    Replaying a learned sequence

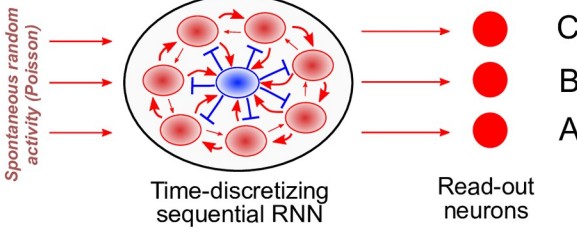

D    Read-out weight matrix after learning

E    Spontaneous sequence reactivations

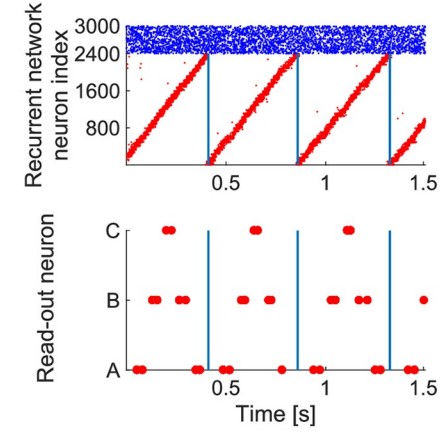

F    Rate of read-out neurons

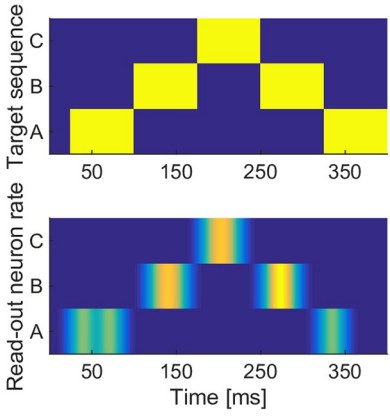

**Fig 3. Learning a non-Markovian sequence via the read-out neurons.** (A) Excitatory neurons in the recurrent network are all-to-all connected to the read-out neurons. The read-out neurons receive additional excitatory input from the supervisor neurons and inhibitory input from interneurons. The supervisor neurons receive spike trains that are drawn from a Poisson process with a rate determined by the target sequence. The read-out synapses are plastic under the voltage-based STDP rule. (B) The rate of the input signal to the supervisor neurons $A$, $B$ and $C$. The supervisor sequence is $ABCBA$ where each letter represents a 75 m$s$ external stimulation of 10 k$Hz$ of the respective supervisor neuron. (C) After learning, the supervisor input and plasticity are turned off. The read-out neurons are now solely driven by the recurrent network. (D) The read-out weight matrix $W^{RE}$ after 12 seconds of learning. (E) Under spontaneous activity, the spikes of recurrent network (top) and read-out (bottom) neurons. Excitatory neurons in the recurrent network reliably drive sequence replays. (F) The target rate (top) and the rate of the read-out neurons (bottom) computed using a one sequence replay and normalized to [0, 1]. The spikes of the read-out neurons are convolved with a Gaussian kernel with a width of $\sim$ 12 m$s$.

preserve the magnitude of the currents in the network, the sparseness of the connectivity ($p$) also varies with $N_C$ such that $pN_C$ is constant. The same training protocols are used for each network configuration as described in the Methods. For a fixed number of clusters $C$, the mean period of the sequential dynamics exhibited by the RNN is largely independent of cluster size $N_C$ (Fig 5A). If we fix the number of neurons in the RNN, and in this way change the

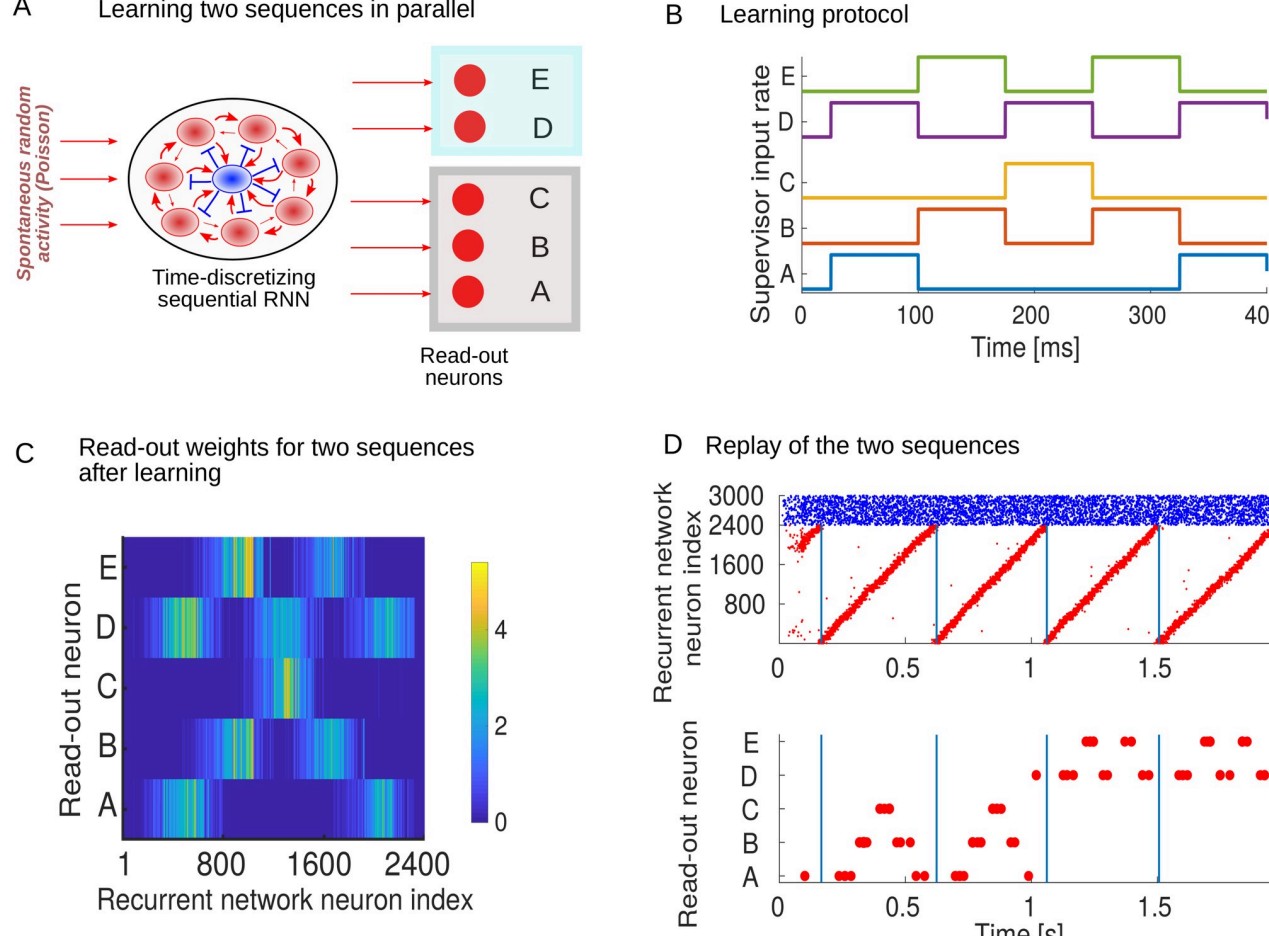

**Fig 4. Learning sequences in parallel.** (A) The recurrent network projects to two sets of neurons. (B) Two different sequences, *ABCBA* and *DEDED*, are learned by alternating between them and presenting each for 2 seconds at a time. (C) The read-out weight matrix after 24 seconds of learning. (D) Raster plot of spontaneous sequence reactivations, where an external inhibitory current is assumed to control which sequence is replayed.

number of clusters $C$, the mean period of the sequential dynamics decreases with increasing cluster size $N_C$. We conclude that the sequential dynamics is preserved over a wide range of network configurations. The time scales in the dynamics depend on the number of clusters and network size.

Another way to modulate the period of the sequential dynamics is to change the unstructured Poisson input to the RNN during spontaneous dynamics (after the first stage of learning). When the rate of the external excitatory input is increased/decreased, the mean period of the sequential dynamics in the RNN decreases/increases (Fig 5B). These results suggest that the network could learn even if the supervisor signal changes in length at each presentation, assuming that both the supervisor and external Poisson input are modulated by the same mechanism.

We next looked at the robustness of the learning of our model under random perturbations and network size. In this context, we consider the effect of cluster size and the deletion of synapses in the read-out layer after learning. We learn the simple *ABCBA* sequence (Fig 3) in the read-out neurons using a RNN with a fixed number of clusters $C$ but varying the cluster size $N_C$. The total learning time ($\Delta_t$) is varied with the cluster size, $N_C \Delta_t$, because smaller clusters

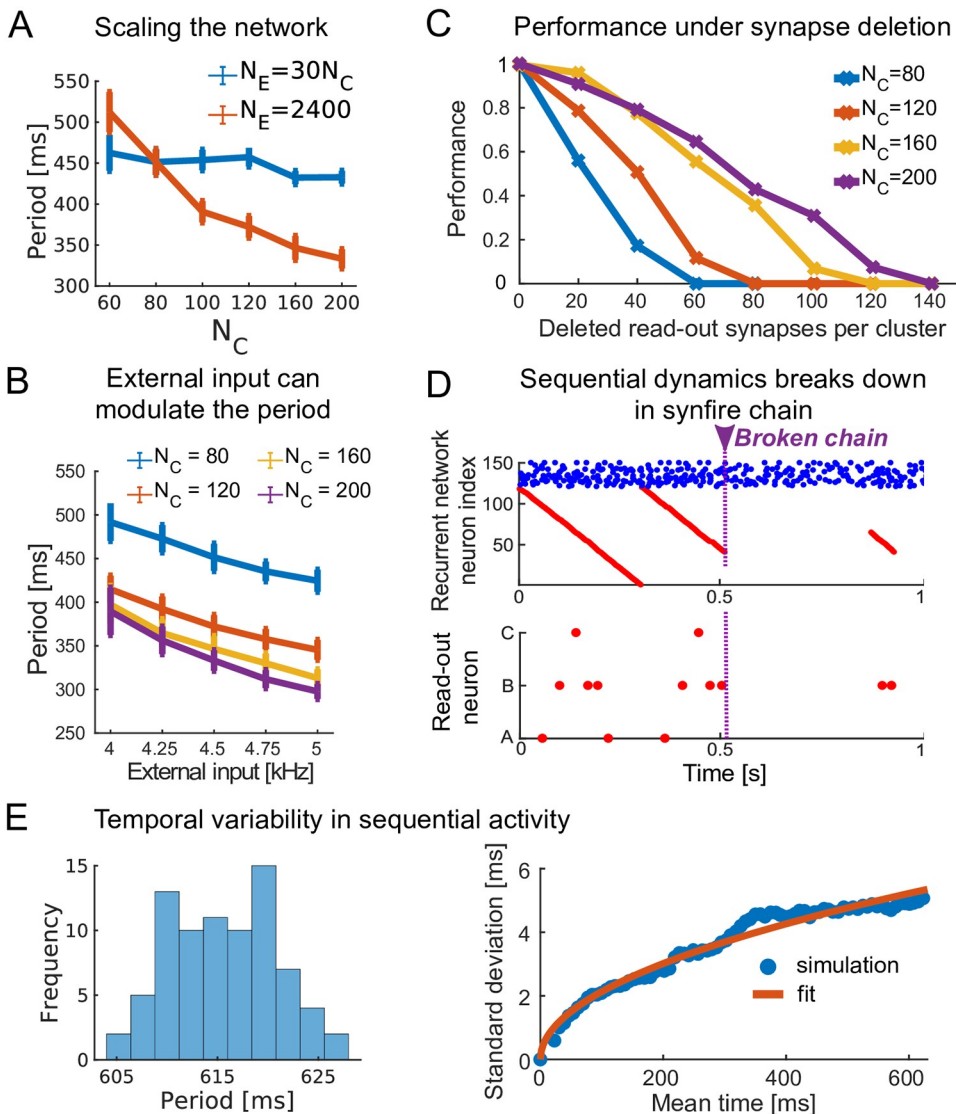

**Fig 5. Scaling, robustness and time variability of the model.** (A) Change of the mean period of the sequential dynamics as the number of clusters grows with: (i) the total number of excitatory neurons kept constant (red line); (ii) the total number of neurons increasing with cluster size (blue line). Error bar shows a standard deviation. (B) Dynamics with varying level of external excitatory input for four different cluster sizes and $N_E = 2400$. The external input can modulate the period of the sequential dynamics by $\sim 10\%$. (C) Recall performance of the learned sequence *ABCBA* for varying cluster sizes and $N_E = 30N_C$ under synapse deletion (computed over 20 repeats). The learning time depends on the cluster size: $\Delta_t = 960s/N_C$. (D) The *ABCBA* sequence is learned with a network of 120 excitatory neurons connected in one large chain and read-out neurons with maximum synaptic read-out strength increased to $W_{max}^{AE} = 75$ pF. The network is driven by a low external input ($r_{ext}^{EE} = 2.75$ kHz). When, at $t = 500$ ms a single synapse is deleted, the dynamics breaks down and parts of the sequence are randomly activated by the external input. Top: spike raster of the excitatory neurons of the RNN. Bottom: spike raster of the read-out neurons. (E) (Left) Histogram of the variability of the period of the sequential activity of the RNN over 79 trials (Right) The standard deviation of the cluster activation time, $\sigma_t$, increases as the square root of $\mu_t$, the mean time of cluster activation: $\sigma_t = 0.213\sqrt{\mu_t}$ (root mean squared error = 0.223 ms).

learn slower, since smaller clusters need larger read-out synaptic strengths to drive the same read-out neuron. We also eliminate an increasing number of synapses in the read-out layer. Performance is quantified as the number of spikes elicited by the read-out neurons after deletion of read-out synapses normalized by the number of spikes elicited before deletion.

Networks with larger clusters show a more robust performance under noise (Fig 5C and S4 Fig). These results show that, not surprisingly, larger clusters drive read-out neurons more robustly and learn faster.

We then tested the limits of time discretization in our model. To that end, we hardcoded a recurrent network with clusters as small as one neuron. In that extreme case, our network becomes a synfire chain with a single neuron in every layer [34]. In this case, randomly removing a synapse in the network will break the sequential dynamics (Fig 5D). Hence, although a spatiotemporal signal can be learned in the read-out neurons, the signal is not stable under a perturbation of the synfire chain. In summary, the choice of cluster size is a trade-off between network size on the one hand and robustness on the other hand. Large clusters: (i) require a large network to produce sequential dynamics with the same period; (ii) are less prone to a failure of the sequential dynamics; (iii) can learn a spatiotemporal signal faster; and (iv) drive the read-out neurons more robustly.

We have also characterized the variability in the duration of the sequential activity, i.e., the period of the RNN. Since the neural activity does not move through the successive clusters with the same speed in each reactivation, we wondered how the variance in the period of our RNN network compared to Weber's law. Weber's law predicts that the standard deviation of reactions in a timing task grows linearly with time [35, 36]. Because time in our RNN is discretized by clusters of neurons that activate each other sequentially, the variability increases over time as in a standard Markov chain diffusive process. Hence the variability of the duration $T$ is expected to grow as $\sqrt{T}$ rather than linearly. This is indeed what our network displays (Fig 5E). Here, we scaled the network up and increased the period of the recurrent network by increasing the network size (80 excitatory clusters of 80 neurons each, see Methods for details).

## Learning a complex sequence

In the non-Markovian target sequence *ABCBA*, the states have the same duration and the same amplitude (Fig 3B). To test whether we could learn more complex sequences, the model was trained using a spatiotemporal signal with components of varying durations and amplitudes. As an example, we use a 'spatio'-temporal signal consisting of a 600 m*s* meadowlark song (Fig 6A). The spectrogram of the sound is normalized and used as the time-varying and amplitude-varying rate of the external Poisson input to the supervisor neurons. Each read-out and supervisor neuron encodes a different frequency range, hence in this example our 'space' dimension is *frequency*.

We first trained a RNN of 6400 excitatory neurons (Fig 5E, see Methods) in order to discretize the time interval spanning the full duration of the song. We then trained the read-out layer. The learned read-out weight matrix reflects the structure of the target sequence (Fig 6B). Under spontaneous activity, the supervisor neurons and interneurons stop firing and the recurrent network drives song replays (Fig 6C). The learned spatiotemporal signal broadly follows the target sequence (Fig 6A). The model performs worse when the target dynamics has time-variations that are faster than or of the same order as the time discretization in the RNN. Thus, we conclude that the model can learn interesting spiking dynamics up to a resolution of time features limited by the time discretization in the recurrent network.

## Discussion

We have proposed here a neuronal network architecture based on biophysically plausible neurons and plasticity rules in order to learn spatiotemporal signals. The architecture is formed by

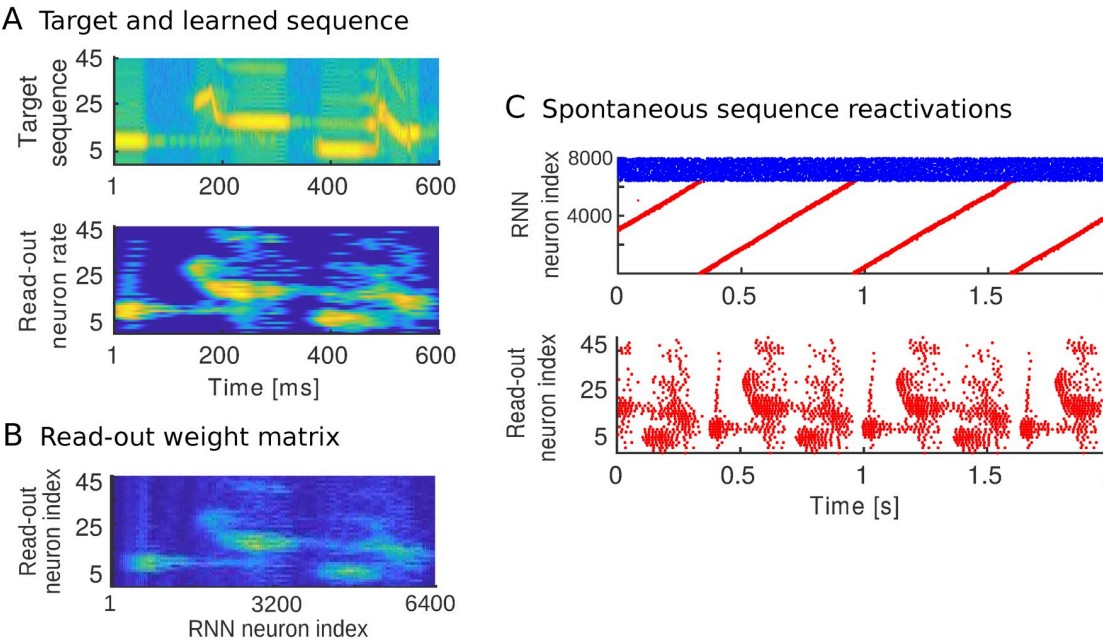

**Fig 6. Learning a complex sequence.** (A) Target sequence (top). The amplitude shows the rate of the Poisson input to the supervisor neurons and is normalized between 0 and 10 k$Hz$. Rate of read-out neurons for one sample reactivation after learning 6 seconds (bottom). 45 read-out neurons encode the different frequencies in the song. Neuron $i$ encodes a frequency interval of $[684 + 171i, 855 + 171i]Hz$. (B) The read-out weight matrix after learning 6 seconds. (C) Sequence replays showing the spike trains of both the recurrent network neurons (top, excitatory neurons in red and inhibitory neurons in blue), and the read-out neurons (bottom).

two modules. The first module provides a temporal backbone that discretizes time, implemented by a recurrent network where excitatory neurons are trained into clusters that become sequentially active due to strong inter-cluster and cluster $i$ to cluster $i + 1$ weights. All of the excitatory clusters are connected to a central cluster of inhibitory neurons. As previously shown for randomly switching clustered dynamics [12], the ongoing spontaneous activity does not degrade the connectivity patterns: the set of plasticity rules and sequential dynamics reinforce each other. This stable sequential dynamics provides a downstream linear decoder with the possibility to read out time at behavioural time scales. The second module is a set of read-out neurons that encode another dimension of a signal, which we generically denote as 'space' but can correspond to any time-varying observable, e.g., spatial coordinates, frequency, discrete states, etc. The read-out neurons learn spike sequences in a supervised manner, and the supervisor sequence is encoded into the read-out weight matrix. Bringing together elements from different studies [18, 22, 32, 37], our model exploits a clock-like dynamics encoded in the RNN to learn a mapping to read-out neurons so as to perform the computational task of learning and replaying spatiotemporal sequences. We illustrated the application of our scheme on a simple non-Markovian state transition sequence, a combination of two such simple sequences, and a time series with more complex dynamics from bird singing.

Other studies have focused on the classification of spatiotemporal signals. The tempotron classifies a spatiotemporal pattern by either producing a spike or not [38]. More recent studies on sequential working memory propose similar model architectures that enable the use of Hebbian plasticity [39]. For example, spatiotemporal input patterns can be encoded in a set of feedforward synapses using STDP-type rules [40, 41]. Combining these approaches with our model might be an interesting line of future research.

The dynamics of the recurrent network spans three time scales: (i) individual neurons fire at the millisecond time scale; (ii) clusters of neurons fire at the "tick of the clock", $\tau_c$, i.e., the time scale that determines the time discretization of our temporal backbone; and (iii) the slowest time scale is at the level of the entire network, i.e. the period of the sequential activity, $\tau_p$, achieved over the cascade of sequential activations of the clusters (see Fig 5A). The time scales $\tau_c$ and $\tau_p$ are dependent on several model parameters: the cluster and network size, the average connection strengths within the clusters, and adaptation. Smaller cluster sizes lead to a smaller $\tau_c$ when the network size is fixed and conversely $\tau_p$ increases with network size when the cluster size is fixed.

The recurrent network is the "engine" that, once established, drives read-out dynamics. Our model can learn different read-out synapses in parallel (Fig 4) and is robust to synapse failure (Fig 5C). This robustness is a consequence of the clustered organization of the recurrent network. Previously proposed models are also robust to similar levels of noise [22, 42]. While an exact comparison is hard to draw, we have shown that it is possible to retain robustness while moving towards a more biological learning rule. The development of a clustered organization in the RNN allows a large drive for the read-out neurons while keeping the individual synaptic strengths reasonably small. If the clusters become small, larger read-out synaptic strengths are required, and the dynamics become less robust. Indeed, the sequential dynamics is especially fragile in the limit where every cluster has exactly one neuron. Furthermore, we show that learning is faster with more neurons per cluster since relatively small changes in the synapses are sufficient to learn the target. This is consistent with the intuitive idea that some redundancy in the network can lead to an increased learning speed [43].

In its current form, the target pattern needs to be presented repeatedly to the network and does not support one-shot learning through a single supervisor presentation. Although increasing the cluster size $N_C$ can reduce the number of presentations, the size of the clusters would need to be impractically large for one-shot learning. Alternatively, increasing the learning rate of the plastic read-out weights could be a way to reduce the number of target presentations to just one [42]. However, it is unclear at present whether such high learning rates are supported by experiments [44].

Taken together, our numerical simulations suggest ways to scale our network. An optimal network configuration can be chosen given the temporal length of the target sequence, and requirements on the temporal precision and robustness. For example, we have shown that a network with $N_E$ = 6400 and $N_C$ = 200 can be chosen for a 400 m$s$ target sequence that can be learned fast with good temporal precision and a robust replay. If the network configuration and size are fixed, this severely constrains the sequences that can be learned and how they are replayed.

In this paper, we use local Hebbian learning to produce a sequential dynamics in the recurrent network. This is in contrast with previous studies, where often a recursive least squares method is used to train the weights of the recurrent network [20, 22, 36, 45]. Hardcoding a weight structure into the recurrent network has been shown to result in a similar sequential dynamics [16, 17, 46]. Studies that do incorporate realistic plasticity rules are mostly focusing on purely feedforward synfire chains [47–49], generating sequential dynamics. In this regard, the contribution of our work is to use the sequential dynamics as a key element to learning spatiotemporal spiking patterns. The ubiquity of sequential dynamics in various brain regions [50–52] and the architecture of the songbird system [53] was an inspiration for the proposed separation of temporal and spatial information in our setup. As we have shown, this separation enables the use of a local Hebbian plasticity rule in the read-out synapses. Our model would therefore predict that perturbing the sequential activity should lead to learning impairments.

Further perturbation experiments can test this idea and shed light on the mechanisms of sequential learning [54].

Previous studies have discussed whether sequences are learned and executed serially or hierarchically [55]. Our recurrent network has a serial organization. When the sequential activity breaks down halfway, the remaining clusters are not activated further. A hierarchical structure would avoid such complete breakdowns at the cost of a more complicated hardware to control the system. Sequences that are chunked in sub-sequences can be learned separately and chained together. When there are errors in early sub-sequences this will less likely affect the later sub-sequences. A hierarchical organization might also improve the capacity of the network. In our proposed serial organization, the number of spatiotemporal patterns that can be stored is equal to the number of read-out neurons. A hierarchical system could be one way to extract general patterns and reduce the number of necessary read-out neurons. Evidence for hierarchical structures is found throughout the literature [56–58]. The basal ganglia is for example thought to play an important role in shaping and controlling action sequences [59–61]. Another reason why a hierarchical organization seems beneficial is inherent to the sequential dynamics. The time-variability of the sequential activity grows by approximately $\sqrt{t}$ (see Fig 5E). While on a time scale of a few hundreds of milliseconds, this does not pose a problem, for longer target sequences this variability would exceed the plasticity time constants. The presented model could thus serve as an elementary building block of a more complex hierarchy.

In summary, we have demonstrated that a clustered network organization can be a powerful substrate for learning, moving biological learning systems closer to machine learning performance. Specifically, the model dissociates temporal and spatial information and therefore can make use of Hebbian plasticity to learn spatiotemporal sequences over behavioural time scales. More general, the backbone as a clustered connectivity might encode any variable $x$ and enable downstream read-out neurons to learn and compute any function of this variable, $\phi(x)$.

## Materials and methods

### Neuron and synapse models

Excitatory neurons are modelled with the adaptive exponential integrate-and-fire model [26]. A classical integrate-and-fire model is used for the inhibitory neurons. All excitatory to excitatory recurrent synapses are plastic under the voltage-based STDP rule [24]. This enables the creation of neuronal clusters and a feedforward structure. Normalization and weight bounds are used to introduce competition and keep the recurrent network stable. Synapses from inhibitory to excitatory neurons in the recurrent network are also plastic under a local plasticity rule [27]. In general, it prevents runaway dynamics and allows for an automatic search of good parameters (S5 Fig). The connections from the recurrent network to the read-out neurons are plastic under the same voltage-based STDP rule. However, potentiation of read-out synapses is linearly dependent on the strength of the synapses. There is no normalization here to allow a continuous weight distribution. The dynamics was chosen based on previous models, with parameters for the dynamics and plasticity to a large extent conserved [12]. More simple integrate-and-fire dynamics should lead to the same qualitative results, given that the parameters are appropriately changed.

### Network dynamics

**Recurrent network.** A network with $N^E$ excitatory ($E$) and $N^I$ inhibitory ($I$) neurons is homogeneously recurrently connected with connection probability $p$. Our network is balanced in terms of inhibition and excitation, so that it displays irregular and asynchronous spiking.

This is signalled by the coefficient of variation (CV) of the inter-spike intervals of the neurons being $CV \sim 1$, thus indicating Poisson-like spiking [62]. In our construction, we initialise the weights of the network near the balanced state by scaling the weights of the balanced RNN in Ref. [12] by the square root of the relative network size (note that this is the reason why some parameters in the table are not round numbers; we never fine-tuned parameters). We then verify that the scaled parameters indeed lead to irregular dynamics. The spiking dynamics is slightly more regular on average, with a mean $CV \sim 0.8$ for excitatory neurons and a mean $CV \sim 0.9$ for inhibitory neurons.

**Read-out neurons.** The $N^E$ excitatory neurons from the recurrent network are all-to-all connected to $N^R$ excitatory read-out (R) neurons. This weight matrix is denoted by $W^{RE}$ and it is where the learned sequence is stored. To help learning, there are two additional types of neurons in the read-out network. During learning, the read-out neurons receive supervisory input from $N^R$ excitatory supervisor (S) neurons. The connection from supervisor neurons to read-out neurons is one-to-one and fixed, $w^{RS}$. Also during learning, $N^R$ interneurons (H) are one-to-one and bidirectionally connected to the read-out neurons with fixed connection strengths, $w^{RH}$ and $w^{HR}$ (see Table 1 for the recurrent network and read-out parameters). The E to E, I to E and the E to R connections are plastic.

**Membrane potential dynamics.** There are two different regimes, one for each part of the model. Excitatory neurons in the recurrent network have a high adaptation current while excitatory neurons in the read-out network have no adaptation. This is to allow for a wide range of firing rates in the read-out network, while spiking is more restricted in the recurrent network. Differences in the refractory period are there for the same reason, but are not crucial. The membrane potential of the excitatory neurons ($V^E$) in the recurrent network has the following dynamics:

$$\frac{dV^E}{dt} = \frac{1}{\tau^E}\left(E_L^E - V^E + \Delta_T^E \exp\left(\frac{V^E - V_T^E}{\Delta_T^E}\right)\right) + g^{EE}\frac{E^E - V^E}{C} + g^{EI}\frac{E^I - V^E}{C} - \frac{a^E}{C} \qquad (1)$$

where $\tau^E$ is the membrane time constant, $E_L^E$ is the reversal potential, $\Delta_T^E$ is the slope of the exponential, $C$ is the capacitance, $g^{EE}$, $g^{EI}$ are synaptic input from excitatory and inhibitory neurons respectively and $E^E$, $E^I$ are the excitatory and inhibitory reversal potentials respectively. When the membrane potential diverges and exceeds 20 mV, the neuron fires a spike and the membrane potential is reset to $V_r$. This reset potential is the same for all neurons in the model. There is an absolute refractory period of $\tau_{abs}$. The parameter $V_T^E$ is adaptive for

**Table 1. Initialization of network.**

| Constant | Value | Description |
|---|---|---|
| $N^E$ | 2400 | Number of recurrent E neurons |
| $N^I$ | 600 | Number of recurrent I neurons |
| $p$ | 0.2 | Recurrent network connection probability |
| $w_0^{EE}$ | 2.83 pF | Initial E to E synaptic strength |
| $w^{IE}$ | 1.96 pF | E to I synaptic strength |
| $w_0^{EI}$ | 62.87 pF | Initial I to E synaptic strength |
| $w^{II}$ | 20.91 pF | I to I synaptic strength |
| $w_0^{RE}$ | 0 pF | Initial E to R synaptic strength |
| $w^{RS}$ | 200 pF | S to R synaptic strength |
| $w^{RH}$ | 200 pF | H to R synaptic strength |
| $w^{HR}$ | 200 pF | R to H synaptic strength |

excitatory neurons and set to $V_T + A_T$ after a spike, relaxing back to $V_T$ with time constant $\tau_T$:

$$\tau_T \frac{dV_T^E}{dt} = V_T - V_T^E. \tag{2}$$

The adaptation current $a^E$ for recurrent excitatory neurons follows:

$$\tau_a \frac{da^E}{dt} = -a^E. \tag{3}$$

where $\tau_a$ is the time constant for the adaptation current (see also S6 Fig). The adaptation current is increased with a constant $\beta$ when the neuron spikes. The membrane potential of the read-out ($V^R$) neurons has no adaptation current:

$$\frac{dV^R}{dt} = \frac{1}{\tau^E} \left( E_L^E - V^R + \Delta_T^E \exp\left( \frac{V^R - V_T^R}{\Delta_T^E} \right) \right) \\ + g^{RE} \frac{E^E - V^R}{C} + g^{RS} \frac{E^E - V^R}{C} + g^{RH} \frac{E^I - V^R}{C} \tag{4}$$

where $\tau^E$, $E_L^E$, $\Delta_T^E$, $E^E$, $E^I$ and $C$ are as defined before. $g^{RE}$ is the excitatory input from the recurrent network. $g^{RS}$ is the excitatory input from the supervisor neuron (supervisor input only non-zero during learning, when the target sequence is repeatedly presented). $g^{RH}$ is the inhibitory input from the interneuron (only non-zero during learning, to have a gradual learning in the read-out synapses). The absolute refractory period is $\tau_{absR}$. The threshold $V_T^R$ follows the same dynamics as $V_T^E$, with the same parameters. The membrane potential of the supervisor neurons ($V^S$) has no inhibitory input and no adaptation current:

$$\frac{dV^S}{dt} = \frac{1}{\tau^E} \left( E_L^E - V^S + \Delta_T^E \exp\left( \frac{V^S - V_T^S}{\Delta_T^E} \right) \right) + g^{SE} \frac{E^E - V^S}{C} \tag{5}$$

where the constant parameters are defined as before and $g^{SE}$ is the external excitatory input from the target sequence. The absolute refractory period is $\tau_{absS}$. The threshold $V_T^S$ follows again the same dynamics as $V_T^E$, with the same parameters. The membrane potential of the inhibitory neurons ($V^I$) in the recurrent network has the following dynamics:

$$\frac{dV^I}{dt} = \frac{E_L^I - V^I}{\tau^I} + g^{IE} \frac{E^E - V^I}{C} + g^{II} \frac{E^I - V^I}{C}. \tag{6}$$

where $\tau^I$ is the inhibitory membrane time constant, $E_L^I$ is the inhibitory reversal potential and $E^E$, $E^I$ are the excitatory and inhibitory resting potentials respectively. $g^{EE}$ and $g^{EI}$ are synaptic input from recurrent excitatory and inhibitory neurons respectively. Inhibitory neurons spike when the membrane potential crosses the threshold $V_T$, which is non-adaptive. After this, there is an absolute refractory period of $\tau_{abs}$. There is no adaptation current. The membrane potential of the interneurons ($V^H$) follow the same dynamics and has the same parameters, but there is no inhibitory input:

$$\frac{dV^H}{dt} = \frac{E_L^I - V^H}{\tau^I} + g^{HE} \frac{E^E - V^H}{C} \tag{7}$$

where the excitatory input $g^{HE}$ comes from both the read-out neuron it is attached to and external input. After the threshold $V_T$ is crossed, the interneuron spikes and an absolute refractory period of $\tau_{absH}$ follows. The interneurons inhibit the read-out neurons stronger when they receive strong inputs from the read-out neurons. This slows the potentiation of the read-out

**Table 2. Neuronal membrane dynamics parameters.**

| Constant | Value | Description |
|---|---|---|
| $\tau_E$ | 20 m$s$ | E membrane potential time constant |
| $\tau_I$ | 20 m$s$ | I membrane potential time constant |
| $\tau_{abs}$ | 5 m$s$ | Refractory period of E and I neurons |
| $\tau_{absR}$ | 1 m$s$ | R neurons refractory period |
| $\tau_{absS}$ | 1 m$s$ | S neurons refractory period |
| $\tau_{absH}$ | 1 m$s$ | H neurons refractory period |
| $E^E$ | 0 m$V$ | Excitatory reversal potential |
| $E^I$ | −75 m$V$ | Inhibitory reversal potential |
| $E_L^E$ | −70 m$V$ | Excitatory resting potential |
| $E_L^I$ | −62 m$V$ | Inhibitory resting potential |
| $V_r$ | −60 m$V$ | Reset potential (for all neurons the same) |
| $C$ | 300 p$F$ | Capacitance |
| $\Delta_T^E$ | 2 m$V$ | Exponential slope |
| $\tau_T$ | 30 m$s$ | Adaptive threshold time constant |
| $V_T$ | −52 m$V$ | Membrane potential threshold |
| $A_T$ | 10 m$V$ | Adaptive threshold increase constant |
| $\tau_a$ | 100 m$s$ | Adaptation current time constant |
| $\beta$ | 1000 p$A$ | Adaptation current increase constant in RNN |

synapses down and keeps the synapses from potentiating exponentially (see Table 2 for the parameters of the membrane dynamics).

**Synaptic dynamics.** The synaptic conductance of a neuron $i$ is time dependent, it is a convolution of a kernel with the total input to the neuron $i$:

$$g_i^{XY}(t) = K^Y(t) * \left( W_{ext}^X s_{i,ext}^X + \sum_j W_{ij}^{XY} s_j^Y(t) \right). \tag{8}$$

where $X$ and $Y$ denote two different neuron types in the model ($E$, $I$, $R$, $S$ or $H$). $K$ is the difference of exponentials kernel:

$$K^Y(t) = \frac{e^{-t/\tau_d^Y} - e^{-t/\tau_r^Y}}{\tau_d^Y - \tau_r^Y},$$

with a decay time $\tau_d$ and a rise time $\tau_r$ dependent only on whether the neuron is excitatory or inhibitory. There is no external inhibitory input to the supervisor and inter- neurons. During spontaneous activity, there is no external inhibitory input to the recurrent network and a fixed excitatory input rate. The external input to the interneurons has a fixed rate during learning as well. The external input to the supervisor neurons is dependent on the specific learning task. There is no external input to the read-out neurons. The externally incoming spike trains $s_{ext}^X$ are generated from a Poisson process with rates $r_{ext}^X$. The externally generated spike trains enter the network through synapses $W_{ext}^X$ (see Table 3 for the parameters of the synaptic dynamics).

## Plasticity

**Excitatory plasticity.** The voltage-based STDP rule is used [24]. The synaptic weight from excitatory neuron $j$ to excitatory neuron $i$ is changed according to the following differential

**Table 3. Synaptic dynamics parameters.**

| Constant | Value | Description |
|---|---|---|
| $\tau_d^E$ | 6 m$s$ | E decay time constant |
| $\tau_r^E$ | 1 m$s$ | E rise time constant |
| $\tau_d^I$ | 2 m$s$ | I rise time constant |
| $\tau_r^I$ | 0.5 m$s$ | I rise time constant |
| $W_{ext}^E$ | 1.6 p$F$ | External input synaptic strength to E neurons |
| $r_{ext}^E$ | 4.5 k$Hz$ | Rate of external input to E neurons |
| $W_{ext}^I$ | 1.52 p$F$ | External input synaptic strength to I neurons |
| $r_{ext}^I$ | 2.25 k$Hz$ | Rate of external input to I neurons |
| $W_{ext}^S$ | 1.6 p$F$ | External input synaptic strength to S neurons |
| $W_{ext}^H$ | 1.6 p$F$ | External input synaptic strength to H neurons |
| $r_{ext}^H$ | 1.0 k$Hz$ | Rate of external input to H neurons |

equation:

$$\frac{dW_{ij}}{dt} = -A_{LTD}\, s_j(t)\, R(u_i(t) - \theta_{LTD}) + A_{LTP}\, x_j(t)\, R(V_i(t) - \theta_{LTP})\, R(v_i(t) - \theta_{LTD}). \tag{9}$$

Here, $A_{LTD}$ and $A_{LTP}$ are the amplitude of depression and potentiation respectively. $\theta_{LTD}$ and $\theta_{LTP}$ are the voltage thresholds to recruit depression and potentiation respectively, as $R(.)$ denotes the linear-rectifying function ($R(x) = 0$ if $x < 0$ and else $R(x) = x$). $V_i$ is the postsynaptic membrane potential, $u_i$ and $v_i$ are low-pass filtered versions of $V_i$, with respectively time constants $\tau_u$ and $\tau_v$ (see also S6 Fig):

$$\tau_u \frac{du_i}{dt} = V_i - u_i \tag{10}$$

$$\tau_v \frac{dv_i}{dt} = V_i - v_i \tag{11}$$

where $s_j$ is the presynaptic spike train and $x_j$ is the low-pass filtered version of $s_j$ with time constant $\tau_x$:

$$\tau_x \frac{dx_j}{dt} = s_j - x_j. \tag{12}$$

Here the time constant $\tau_x$ is dependent on whether learning happens inside ($E$ to $E$) or outside ($E$ to $R$) the recurrent network. $s_j(t) = 1$ if neuron $j$ spikes at time $t$ and zero otherwise. Competition between synapses in the recurrent network is enforced by a hard $L1$ normalization every 20 m$s$, keeping the sum of all weights onto a neuron constant: $\Sigma_j\, W_{ij} = K$. $E$ to $E$ weights have a lower and upper bound $[W_{min}^{EE}, W_{max}^{EE}]$. The minimum and maximum strengths are important parameters and determine the position of the dominant eigenvalues of $W$. Potentiation of the read-out synapses is weight dependent. Assuming that stronger synapses are harder to potentiate [63], $A_{LTP}$ reduces linearly with $W^{RE}$:

$$A_{LTP} = A\, \frac{W_{max}^{RE} - W^{RE}}{W_{max}^{RE} - W_{min}^{RE}}. \tag{13}$$

The maximum LTP amplitude $A$ is reached when $W^{RE} = W_{min}^{RE}$ (see Table 4 for the parameters of the excitatory plasticity rule).

**Table 4. Excitatory plasticity parameters.**

| Constant | Value | Description |
|---|---|---|
| $A_{LTD}$ | 0.0014 pA mV$^{-2}$ | LTD amplitude |
| $A$ | 0.0008 pA mV$^{-1}$ | LTP amplitude (in RNN: $A_{LTP} = A$) |
| $\theta_{LTD}$ | $-70$ mV | LTD threshold |
| $\theta_{LTP}$ | $-49$ mV | LTP threshold |
| $\tau_u$ | 10 ms | Time constant of low pass filtered postsynaptic membrane potential (LTD) |
| $\tau_v$ | 7 ms | Time constant of low pass filtered postsynaptic membrane potential (LTP) |
| $\tau_{xEE}$ | 3.5 ms | Time constant of low pass filtered presynaptic spike train in recurrent network |
| $\tau_{xRE}$ | 5 ms | Time constant of low pass filtered presynaptic spike train for read-out synapses |
| $W_{min}^{EE}$ | 1.45 pF | Minimum E to E weight |
| $W_{max}^{EE}$ | 32.68 pF | Maximum E to E weight |
| $W_{min}^{RE}$ | 0 pF | Minimum E to R weight |
| $W_{max}^{RE}$ | 25 pF | Maximum E to R weight |

**Inhibitory plasticity.** Inhibitory plasticity acts as a homeostatic mechanism, previously shown to prevent runaway dynamics [12, 13, 27]. Here, it allows to automatically find good parameters (see also S5 Fig). Excitatory neurons that fire with a higher frequency will receive more inhibition. The I to E weights are changed when the presynaptic inhibitory neuron or the postsynaptic excitatory neuron fires [27]:

$$\frac{dW_{ij}}{dt} = A_{inh}\left(y_i^E(t) - 2r_0\tau_y\right) s_j^I(t) + A_{inh} y_j^I(t) s_i^E(t) \tag{14}$$

where $r_0$ is a constant target rate for the postsynaptic excitatory neuron. $s^E$ and $s^I$ are the spike trains of the postsynaptic E and presynaptic I neuron respectively. The spike trains are low pass filtered with time constant $\tau_y$ to obtain $y^E$ and $y^I$ (as in Eq 12). Table 5 shows parameter values for the inhibitory plasticity rule. The I to E synapses have a lower and upper bound $\left[W_{min}^{EI}, W_{max}^{EI}\right]$.

## Learning protocol

Learning happens in two stages. First a sequential dynamics is learned in the RNN. Once this temporal backbone is established connections to read-out neurons can be learned. Read-out neurons are not interconnected and can learn in parallel.

**Recurrent network.** The network is divided in 30 disjoint clusters of 80 neurons. The clusters are sequentially stimulated for a time duration of 60 minutes by a large external current where externally incoming spikes are drawn from a Poisson process with rate 18 kHz. This high input rate does not originate from a single external neuron but rather assumes a large external input population. Each cluster is stimulated for 10 ms and in between cluster

**Table 5. Inhibitory plasticity parameters.**

| Constant | Value | Description |
|---|---|---|
| $A_{inh}$ | $10^{-5}$ AHz | Amplitude of inhibitory plasticity |
| $r_0$ | 3 Hz | Target firing rate |
| $\tau_y$ | 20 ms | Time constant of low pass filtered spike train |
| $W_{min}^{EI}$ | 48.7 pF | Minimum I to E weight |
| $W_{max}^{EI}$ | 243 pF | Maximum I to E weight |

stimulations there are 5 m$s$ gaps (see also S6 Fig for different gaps). During excitatory stimulation of a cluster, all other clusters receive an external inhibitory input with rate 4.5 k$Hz$ and external input weight $W^I_{ext} = 2.4$ p$F$. There is a periodic boundary condition, i.e. after the last cluster is activated, the first cluster is activated again. After the sequential stimulation, the network is spontaneously active for 60 minutes. The connectivity stabilizes during the spontaneous dynamics. Learning in scaled versions of this network happens in exactly the same way (Fig 5A). The recurrent weight matrix of the large network (80 clusters of 80 neurons, Figs 5E and 6) is learned using the same protocol. The recurrent weight matrix reaches a stable structure after three hours of sequential stimulation followed by three hours of spontaneous dynamics. Parameters that change for the scaled up version are summarized in Table 6. For randomly switching dynamics, a similar protocol is followed (S1 Fig). The weight matrix used to plot the spectrum of the recurrent network in Fig 2 and S2 Fig is:

$$W = \begin{pmatrix} W^{EE} & -W^{EI} \\ W^{IE} & -W^{II} \end{pmatrix}.$$

**Read-out network.** During learning of the read-out synapses, external input drives the supervisor and interneurons. The rate of the external Poisson input to the supervisor neurons reflects the sequence that has to be learned. The rate is normalized to be between 0 k$Hz$ and 10 k$Hz$. During learning, $W^{RE}$ changes. After learning, the external input to the supervisor and inter- neurons is turned off and both stop firing. The read-out neurons are now solely driven by the recurrent network. Plasticity is frozen in the read-out synapses after learning. With plasticity on during spontaneous dynamics, the read-out synapses would continue to potentiate because of the coactivation of clusters in the recurrent network and read-out neurons. This would lead to read-out synapses that are all saturated at the upper weight bound.

**Simulations.** The code used for the training and simulation of the recurrent network is built on top of the code from [12] in Julia. The code used for learning spatiotemporal sequences using read-out neurons is written in Matlab. Forward Euler discretization with a time step of 0.1 m$s$ is used. The code is available online on ModelDB (http://modeldb.yale.edu/257609).

**Table 6. Parameters for the large recurrent network (all the other parameters are the same as the smaller network).**

| Constant | Value | Description |
|---|---|---|
| $N^E$ | 6400 | Number of recurrent E neurons |
| $N^I$ | 1600 | Number of recurrent I neurons |
| $w^{EE}_0$ | 1.73 p$F$ | baseline E to E synaptic strength |
| $w^{IE}$ | 1.20 p$F$ | E to I synaptic strength |
| $w^{EI}_0$ | 40 p$F$ | Initial I to E synaptic strength |
| $w^{II}$ | 12.80 p$F$ | I to I synaptic strength |
| $W^{EE}_{min}$ | 1.27 p$F$ | Minimum E to E weight |
| $W^{EE}_{max}$ | 30.5 p$F$ | Maximum E to E weight |
| $W^{EI}_{min}$ | 40 p$F$ | Minimum I to E weight |
| $W^{EI}_{max}$ | 200 p$F$ | Maximum I to E weight |
| $W^{RE}_{max}$ | 15 p$F$ | Maximum E to R weight |

## Linear rate model: Spectral analysis

A linear rate model can give insight into the dynamics of a large nonlinear structured spiking network [15]. The dynamics of a simplified rate model with the same feedforward structure as in the RNN is as follows:

$$\frac{dx}{dt} = -x + Ax + \xi \tag{15}$$

where $x$ is a multidimensional variable consisting of the rates of all excitatory and inhibitory clusters, $A$ is the weight matrix, and $\xi$ is white noise. The matrix $A$ is a coarse-grained version of the weight matrix of the recurrent network in Fig 2B averaged over each cluster. In order to obtain analytical expressions, we consider a network with 3 excitatory clusters and 1 inhibitory cluster. The connectivity of this model can be parametrized as:

$$A = \begin{bmatrix} \delta & 1 & \epsilon & -kw \\ \epsilon & \delta & 1 & -kw \\ 1 & \epsilon & \delta & -kw \\ \frac{w}{3} & \frac{w}{3} & \frac{w}{3} & -kw \end{bmatrix} \tag{16}$$

where $\delta > 0$, $w = \delta + \epsilon + 1$, $\epsilon > 1$ guarantees sequential dynamics, and $k > 1$ guarantees a balanced network.

The Schur decomposition $A = UTU^T$ gives eigenvalues and Schur vectors $\mathbf{u}_i$:

$$T = \begin{bmatrix} -w(k-1) & \sqrt{3}w\left(k+\frac{1}{3}\right) & 0 & 0 \\ 0 & 0 & 0 & 0 \\ 0 & 0 & \delta - \frac{\epsilon+1}{2} & -\frac{\sqrt{3}}{2}(\epsilon-1) \\ 0 & 0 & \frac{\sqrt{3}}{2}(\epsilon-1) & \delta - \frac{\epsilon+1}{2} \end{bmatrix} \tag{17}$$

$$\mathbf{u}_1 = \frac{1}{2}\begin{bmatrix} 1 \\ 1 \\ 1 \\ 1 \end{bmatrix}, \quad \mathbf{u}_2 = \frac{1}{2\sqrt{3}}\begin{bmatrix} 1 \\ 1 \\ 1 \\ -3 \end{bmatrix}, \quad \mathbf{u}_3 = \frac{1}{\sqrt{2}}\begin{bmatrix} -1 \\ 0 \\ 1 \\ 0 \end{bmatrix}, \quad \mathbf{u}_4 = \frac{1}{\sqrt{6}}\begin{bmatrix} -1 \\ 2 \\ -1 \\ 0 \end{bmatrix}. \tag{18}$$

The first mode $\mathbf{u}_1$ decays fast and uniformly over the four neuronal groups. The second mode $\mathbf{u}_2$ decays more slowly, and indicates the interplay between excitatory groups $(x_1, x_2, x_3)$ and the inhibitory group $x_4$. The eigenspace associated with the pair $\{\mathbf{u}_3, \mathbf{u}_4\}$ has complex conjugate eigenvalues and is localized on the three excitatory groups. An increase of activity in one excitatory group is coupled with decreased activities in the other groups. If the real part $\delta - \frac{\epsilon+1}{2} < 1$ then these modes are linearly stable but if the real part becomes closer to one means a slower decay of this mode. Importantly, the imaginary part of the eigenvalues is $\pm\sqrt{3}(\epsilon-1)/2$; hence it grows linearly with the strength of the feedforward structure $(\epsilon - 1)$

**A** Cartoon of linear rate model **B** Eigenvalues of linear rate model

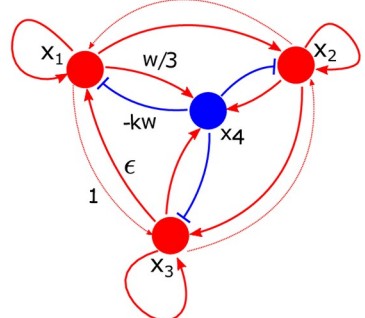 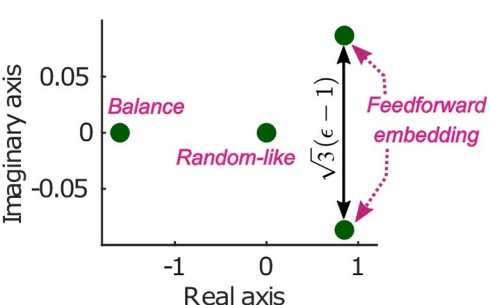

**Fig 7. Spectral analysis of reduced linear model.** (A) Cartoon of a simplified linearised rate model with three nodes $x_1, x_2, x_3$ corresponding to three clusters of excitatory neurons with recurrent strength $\delta$ connected to a central cluster of inhibitory neurons $x_4$. The cyclic connections are stronger clockwise than anticlockwise since $\epsilon > 1$. (B) The spectrum shows a conjugate complex eigenvalue pair with large real part $(2\delta - \epsilon - 1)/2$ and an imaginary part $\pm\sqrt{3}(\epsilon - 1)/2$ which grows linearly with the asymmetry of the clockwise/anticlockwise strength $(\epsilon - 1)$. This pair of eigenvalues dominates the dynamics as their real parts are close to 1 and leads to the periodic behaviour corresponding to propagation around the cycle $x_1 \rightarrow x_2 \rightarrow x_3 \rightarrow x_1 \ldots$.

(Fig 7A). This leads to oscillatory behavior, which determines the time scale of the sequential switching.

## Supporting information

**S1 Fig. Randomly switching dynamics.** The recurrent network is stimulated with external input that is spatially clustered, but temporally uncorrelated. Each cluster is stimulated for 50 m*s*, with 50 m*s* gaps in between stimulations. The rate of external stimulation is 18 k*Hz* during training. This is repeated for 20 minutes after which the network stabilizes during 20 minutes of spontaneous activity. (A) A diagonal structure is learned in the recurrent weight matrix. Since there are no temporal correlations in the external input, there is no off-diagonal structure. (B) The spectrum shows an eigenvalue gap. This indicates the emergence of a slower time scale. The leading eigenvalues do not have an imaginary part, pointing at the absence of feedforward structure and thus there is no sequential dynamics. (C) Under a regime of spontaneous dynamics (i.e. uncorrelated Poisson inputs), the clusters are randomly reactivated. (TIF)

**S2 Fig. The connectivity structure is stable under spontaneous dynamics.** (A) After 60 minutes of training, the network stabilizes during spontaneous activity. During the first 30 minutes of spontaneous dynamics, the connectivity still changes. More specifically, the imaginary parts of the leading eigenvalues increase. This leads to a higher switching frequency and as such a smaller period in the sequential activity. After around 30 minutes, a fixed point is reached. The first row shows spike trains at different times, for one second of spontaneous activity. The second row shows the spectra of the weight matrix at those times. (B) After 60 minutes of sequential stimulation, we test reinforcement and degradation of the learned connectivity by decoupling the plasticity from the dynamics. We plot the evolution of the off-diagonal weights during spontaneous dynamics in two separate cases: (i) we run the dynamics of the network using a frozen copy of the learned weight matrix and apply plastic changes that result from the dynamics to the original weight matrix (blue curve); (ii) we run the dynamics of the network using a frozen copy of the learned weight matrix where the off-diagonal structure was removed and apply plastic changes that result from the dynamics to the original weight matrix (red

curve). We can see that in the former, the learned connectivity is reinforced and in the latter, the learned connectivity degrades. Off-diagonal weights (the y-axis) are quantified by averaging over the weights in the 80 by 80 blocks in the lower diagonal, for the 30 different clusters. The curves are the means over the 30 clusters and the error bars one standard deviation.
(TIF)

**S3 Fig. Noisy learning.** The sequence *ABCBA* is relearned four times for 12 seconds each. Before relearning, the read-out weight matrix $W^{RE}$ was always reset. When active, read-out neurons fire two spikes on average +/− one spike. This variability is a consequence of the noisy learning process.
(TIF)

**S4 Fig. Details of some network properties.** (A) Duration that a cluster is activated as a function of network size (B) Raster plot of sequential dynamics for $N_E = 1200$ and $N_C = 40$, after training. We observe that by reducing the cluster size, the irregularities in the sequential dynamics are increased (compare with Fig 2). (C) Two raster plots showing two different levels of robustness (summary plot in Fig 5C). In both cases, at $t = 1s$ (purple arrow), 40 read-out synapses are deleted for each cluster. Left panel: $N_C = 120$, each read-out neuron fires two spikes before deletion and one spike after deletion resulting in $\sim$ 50% performance. Right panel: $N_C = 200$, each read-out neuron fires two spikes before deletion and one or two spikes after deletion resulting in a higher performance ($\sim$ 80%).
(TIF)

**S5 Fig. The role of inhibition.** (A) Inhibitory neurons are necessary to prevent pathological excitatory activity. (B) The weights projecting from the inhibitory neurons to the excitatory neurons without inhibitory plasticity are random (left panel). The weights projecting from the inhibitory neurons to the excitatory neurons with inhibitory plasticity show some structure (right panel). (C) The full spectrum of the recurrent weight matrix after learning without inhibitory plasticity. (D) Without inhibitory plasticity, the sequential dynamics shows irregularities. The inhibitory plasticity allows for better parameters to be found to stabilize the sequential dynamics in the recurrent network.
(TIF)

**S6 Fig. Sensitivity to parameters.** The periods of the sequential dynamics are computed after one hour of external stimulation and one hour of spontaneous dynamics. Only one parameter at a time is changed. (A) The adaptation time constant is varied. (B) The time gap between external sequential stimulations is varied. (C) The time constants of the voltage-based STDP rule are varied. The lines are guides to the eye and the error bars indicate one standard deviation.
(TIF)

## Author Contributions

**Conceptualization:** Amadeus Maes, Mauricio Barahona, Claudia Clopath.

**Formal analysis:** Amadeus Maes, Mauricio Barahona, Claudia Clopath.

**Investigation:** Amadeus Maes, Mauricio Barahona, Claudia Clopath.

**Methodology:** Amadeus Maes.

**Supervision:** Mauricio Barahona, Claudia Clopath.

**Visualization:** Amadeus Maes.

**Writing – original draft:** Amadeus Maes.

**Writing – review & editing:** Amadeus Maes, Mauricio Barahona, Claudia Clopath.

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
