## [Decision Letter · Decision Letter 0]

26 Aug 2019

Dear Dr Clopath,

Thank you very much for submitting your manuscript 'Learning spatiotemporal signals using a recurrent spiking network that discretizes time' for review by PLOS Computational Biology. Your manuscript has been fully evaluated by the PLOS Computational Biology editorial team and in this case also by independent peer reviewers. The reviewers appreciated the attention to an important problem, but raised some substantial concerns about the manuscript as it currently stands. While your manuscript cannot be accepted in its present form, we are willing to consider a revised version in which the issues raised by the reviewers have been adequately addressed. We cannot, of course, promise publication at that time. 

Both reviewers have identified questions of robustness as issues in the paper. The reviewers asked a number of questions this way, regarding matching of time constants to stimulus speed, importance of specific transmission delays, network size, synapse number, etc. It is important that you make clear how the model operates in a variety of regimes, per the reviewers' requests. Reviewer 1 has also given you a number of detailed comments on the writing you should attend to. And, we would also encourage you to fully address Reviewer 2's question about how this relates to other RNN models, including the more biologically plausible ones the reviewer cites.

Sincerely,

Blake A. Richards

Associate Editor

PLOS Computational Biology

Kim Blackwell

Deputy Editor

PLOS Computational Biology

[LINK]

Reviewer's Responses to Questions

**Comments to the Authors:**

Reviewer #1: Dear authors,

I originally wrote a review that was 28250 characters long, but PLOS only accepts reviews up to 20000 characters long. The only way to get to this length was to remove all praise, all encouragement, and all hedging words ("in my opinion", "Why would not you try" etc.), making the review sound angry! So please see this angry version below :) As the review is signed, I'll also send you, and separately the editor, a human-readable version by email (for some of the changes, it also has some additional rationale, and some ideas on how to make these changes). Thanks!

A.

-------------

Your paper "Learning spatiotemporal signals using a recurrent spiking network that discretizes time" is an interesting and promising piece of work. You show that a simple architecture, consisting of a self-organizing recurrent network and an output layer, and trained through a form of 2-stage curriculum learning, can learn to reproduce complex temporal patterns that are much slower than the time scale at which individual neurons operate. This is a neat illustration of what a relatively simple model can do, and I believe it can become a foundation for a very interesting line of research!

The current version of the paper suffers from two weaknesses:

First, please revise your writing in several ways (see below for details)

Second, the paper introduces several special fine-tuned cases, and demonstrates that they work or don't work, but does not offer any conceptual sensitivity analysis. The paper will be stronger if you finish some (or ideally, all) of these lines of inquiry with proper sensitivity analyses. I recommend:

1) Based on Fig2D, show how the quality of final stimulus replay (or any other measure of "success" for this model and this type of learning) depends on the exact pairing between the stimulus phase speed (100 ms) and the adaptation current time constant tau_a (100 ms). From Fig2D it seems that the stimulus is reproduced by the network so well largely because the neurons are "hard-tuned" to inactivate at exactly the speed you are using for stimulus transitions. It is not necessarily a problem, but it is an important aspect of conceptualizing the model. You could keep the stimulus fixed, change tau_a, and plot the replay_quality(tau_a).

2) For the comparison of large and small networks, make your statements more clear. The two plots shown in Figure 4A are practically identical, yet you make statements about these two models (large and small) both in the figure caption, and in the text. If there are differences there, in terms of time required for training, and/or their robustness, please show these numbers on the figure.

3) The most interesting analysis, in my opinion, would be one to explore the "optimal" number of clusters (and thus, the size of each cluster), for a model of N neurons, given a certain level of stimulus complexity, and a certain amount of noise (or neuron silencing). If for a given pair of (complexity, noise) you can _predict_ the optimal network configuration, this would be a really interesting result, and it would make the paper more memorable, and impactful. Below I sketch how it can be done.

# Detailed comments

Page 1

"slow-switching dynamics [Schaub et al., 2015]" - Please clarify

Later in the same sentence: "and different types of sequential dynamics" - what are the options here? Can they be succinctly summarized?

"use non-local information either in space or in time" - explain these terms to non-computational readers

Page 2

"All synapses are plastic under typical Hebbian learning rules" - is STDP considered a typical Hebbian learning rule? Please be clear about whether you use a simple Hebbian potentiation, or a more fancy rule, such as STDP.

"Thus, the recurrent network encodes time and the read-out neurons encode ’space’." - I am not sure I can understand this sentence. Could you please restate it, making it more clear?

"Similar to the liquid state-machine" - remind the reader what a liquid state-machine is.

"Time is discretized in the recurrent spiking network by C clusters of excitatory neurons"

1) How time can be discretized by a cluster of neurons? Isn't time discretization a feature of your model, i.e. a step at which you perform your calculations? After re-reading the paper 2 times, I think I know what you mean, but I was very lost the first time I read it. Please clarify.

2) mention what general type of a model you use for your neurons. (something like a "differential equation with 10 global parameters and 7 hidden variables per neuron, and a hard reset for spiking") If there a short reference for your model, cite it

3) introduce the concept of clusters separately. At this point I was really confused whether those 'clusters' you mentioned were just a metaphor, or whether they were a defining feature of your network. I also wondered whether they arose naturally, through the dynamics of activation, or whether they were introduced in the network deliberately.

"This means that at each time only a subset of neurons spike, i.d. the neurons belonging to the same cluster."

1) What does "this" refer to? Please clarify.

2) Why does the previous sentence mean ("this means") that at each time only a subset of neurons spike? Why cannot all of them spike at once?

3) I don't know what "i.d." means, and I could not immediately google it, which may suggest that it is not a widely used abbreviation.

"At time t in [t_{i-1}; t_i] the read-out neurons are solely driven by the neurons in cluster i, with some additional background noise. This discretization enables Hebbian plasticity to bind the read-out neurons to the neurons active in the relevant time-bin." - this part is really unclear. Please rewrite and clarify.

"Here however, we use the voltage-based STDP plasticity rule at the excitatory to excitatory neurons" - is it true for all connections in the model, or only for some of them?

"In the first stage, a feedforward weight structure is embedded into the recurrent network." - what does the word "embedded" mean in this context? Does it just mean that you initialize the network with some feed-forward way, or does it have a deeper meeting? While reading, at this point I assumed that you were describing some peculiar initial state, but without going to the "Methods" section, I was not sure whether my guess was correct. Please try to explain: 1) why you are trying to do something, 2) what is it exactly that you're trying to do, 3) how you're doing it.

"The recurrent network is initialized as a balanced network with random connectivity."

1) You just said that you "embedded" some FF connectivity in it, and now you are saying the connectivity is random. At the first pass, it read as a contradiction.

2) Please clarify what "balanced" means in this context.

"To embed a feedforward structure, the network is stimulated in a sequential manner" - this return to a topic you abandoned 2 sentences ago makes it harder to stay on track.

"(rate of 18 kHz for 10 ms)." - you have not yet mentioned the time step of your calculations; it may be better to mention it at some point before this sentence, to put these numbers in perspective. But also, I am confused about this section, as in the very end of Methods section on page 14 you are saying that you worked with a time step of 0.1 ms. Would not 5 kHz be the highest possible frequency of spiking achievable in a model that is discretized at 10 kHz? How can you use 18 kHz then?

"During the stimulation, neurons in the same cluster fire spikes together, which will strengthen the intra-cluster connections bidirectionally through the voltage-based STDP rule"

1) This sentence and 2-3 sentences before it are intriguing, but at this point the reader does not yet know _why_ you do what you do. Is it the experiment already? Or are you trying to build a certain architecture using something like a curriculum learning, through inputs alone? Please make sure to describe what you are trying to achieve, and why you are trying to achieve it, before describing how you do it.

2) I always thought that the "canonical" STPD does not link neurons that spike together, but rather potentiates synapses connecting neurons that fire with a small delay of several ms. With 0.1 ms step and high-frequency stimulation, this difference could be very noticeable. Yet here you say that synchronous firing strengthened within-cluster connections bidirectionally. Is it because your STDP is symmetric? Or is it because neurons fire so much that they develop tight synfire chains within each cluster? Please clarify.

"When the time gap between sequential stimulations is increased during the training phase, there is no pre/post pairing between clusters anymore." - this sounds like the first true "Result", but I am still not sure what question you are trying to answer. Stating a question upfront would help a lot.

"This leads to slow-switching dynamics as opposed to sequential dynamics" - please clarify 1) what the word "this" refers to, and 2) what is the "slow-switching" dynamics, and how is it different from "sequential dynamics". You have a whole figure on it (Figure S1), so it would be really useful to define this term here, with a few references, and explain what this result means.

"The connectivity pattern is therefore stable" - from the Abstract, I remember that overall you found this result intersting. If it is true, please expand your writing a bit, and explain why this is an interesting statement. Were you concerned that the dynamics would degrade? Are there previous studies that suggest the risk of degradation? If so, reference these studies, and explain your motivation, to properly frame this result.

"The feedforward weight embedding changes the spectrum of the recurrent weight matrix." - This sounds like a beginning of a new thought, which clearly deserves a new paragraph. I would also strongly suggest that you start this paragraph with 2-3 sentences, explaining what is a spectrum of a recurrent weight matrix, and how to interpret it (with 1-2 references). Make it easier for non-mathy (or intermediate-mathy) people to relate to your work.

Figure 2C (The drawing with an ellipse): While I know the idea of spectral analysis, I don't work with it daily, and with my level of background, this panel is completely impenetrable for me. I don't know how to read it, and what it means. I think the paper would much improve if this section about eigenvalues, and the Figure 2, are better referenced, and better explained.

"Analysis of a simplified linear rate model shows that the imaginary parts linearly depend on the strength of the feedforward embedding" - can you please expand on that, including why you did it, how you did it, and what was the result? Ideally, frame it as a sequence of simple statements. There's a known method. Here's how it works. In our case, conceptually, we could have expected to see this or that (which would have meant this or that). In practice, we observed this. Some aspect of it was surprising (what exactly), so we ran this additional model (why and how). We observed this. Here's what it means.

"A large adaptation current counteracts the recurrent reverberating activity to turn the activity reliably off." - but so, does it mean that the temporal dynamics or your spontaneous activity was not due to training, but due to this "hard-coded" adaptation in individual neurons? (see my next paragraph)

Page 4

"Therefore, as each cluster is spontaneously active in a sequence, that leads to a sequence length that reaches behavioural time scales" - So, are you saying that to jump from one cluster to another (say, from i to i+1), the activation in the cluster i should have reached a certain threshold, but at the same time the adaptation was driving this activation down? Does it mean that with adaptation too fast or too strong, there would be no propagation one cluster to another, while with adaptation too slow or too weak, clusters would not have "switched off"? But if so, would not your model work only if the adaptation is precisely tuned to the time parameter of your stimulus? This may actually be my biggest question about this study, as whether this model generalizes well seems to hinge on it. If this architecture can work successfully only when adaptation at the level of individual cells is precisely tuned to the properties of the stimulus encoded by the entire network, this could be a strong limitation (in the spirit of non-local tuning). If however this architecture is robust, and can somehow tolerate a large range of adaptation properties, it would make it a good candidate for explaining encoding of temporal stimuli in the brain. The answer to this question is also tightly linked to some other questions raised in the paper: most importantly, the link between the number of clusters, the coding capacity of the network, and its robustness to noise (something I discuss more below). In order to relate to any analyses that play with the number of clusters in the network (in order to conceptually generalize them) the reader would need to know how these analyses would change if the adaptation time scale is not carefully pre-selected. If the quality of propagation is not too sensitive to the adaptation time, we need to know that. If, on the other extreme, the adaptation time sets the speed of this propagation, or needs to be strongly coordinated with some variables that set the speed of this propagation, we should know that as well, and keep it in mind, when interpreting any conceptual predictions from the model.

"simple yet non-markovian sequence" - please define what a markovian or non-markovian sequence means in this context, with a reference.

"This task is non-trivial and it is unclear how to solve this problem in a generic recurrent spiking network with local plasticity rules." - this sentence would benefit from a bit of stylistic re-write, and also it clearly asks for a reference or two.

"However, separating time and space solves this in a natural way." - very interesting, but also very unclear. It is the second time this theme is introduced, and I still don't understand it. If you feel strongly about it, please clarify what you mean, and try to explain to the reader why they should be excited about it (essentially, help them to understand how it may help them to conceptualize their own work later on).

"Our recurrent network (Fig 2) is used here, where the first cluster is activated at time t." - This sentence would benefit from a rewrite.

"Due to the voltage-based plasticity, synapses are potentiated from neurons in the recurrent network and read-out neurons that fire at the same time"

1) I am not sure you mentioned before that these synapses are special. Please introduce it better.

2) Please explain how this special "voltage-based plasticity" helps here, compared to "more standard" Hebbian plasticity the reader may be thinking about.

"To that end, we trained our recurrent network with clusters as small as one neuron." - Personally, I find this line of inquiry very interesting, as it speaks of brain architecture optimization, in terms of robustness to perturbation. However, you don't really explore this question, but look at one extreme case, and then make a prediction based on this extreme case: "In summary, the choice of cluster size is a trade-off between network size on the one hand and robustness on the other hand." I wonder whether a proper analysis of this relationship could be one of the main points of the paper, if run properly. What is the optimal number of clusters, for a stimulus of given complexity (say, a number of unique state transitions, or even just a number of states for a randomized sequence), and given amount of noise / perturbation?

Page 9

"These clusters are sequentially active by embedding a feedforward connectivity structure into the weights of the network." - the semantics is off here. If you think of it, clusters cannot be active by embedding. Please rewrite.

"The time scales tau_c and tau_p are dependent on the cluster and network size, the average connection strengths within

the clusters and adaptation. Smaller cluster sizes lead to a smaller tau_c and tau_p increases with network size when

the cluster size is fixed." - arguably you don't address it in the paper, but only hint at it. To put it in a slightly exaggerated way, there is no summary plot among your figures that would have cluster size as an X axis, and some time-related measure as an Y axis. Or, time-related measure as X, encoding quality as Y, and several (more than 2) cluster sizes as different lines. If you don't think these summary analyses belong here, please restate your statements as hypotheses. If however you'd like to keep them as statements, please run appropriate analyses, and visualize them in one of the figures (I would definitely advocate for this second approach!)

METHODS

Pages 10-11

"Membrane potential dynamics" - as I mentioned before, I am not quite sure how to classify your model, and this entire section does not contain any references. Your model seems to be simpler than HH, but more fancy than Izhikevich, and you are using a computationally expensive exp() function. The obvious question is: why? Would not integrate-and-fire be enough? Or a classic Izhikevich neuron? Are you using this model just because that was the first model you thought of, or do you NEED some of its properties to make your model work? I don't necessarily suggest that you run a simulation with different types of neurons, but if you know that you had to reach this level of complexity, perhaps because of some prior pilot experiments, I would really want you to write about that. And even if you use this model just because it was a reasonable "first choice", I would like to know about it, to avoid second-guessing.

To rephrase, I would be grateful if you put your model in a context (with some references), and commented on your choice for the level of complexity in modeling your neurons. A few choices that jump at me as I read these section are your attention to refractory periods and adaptation currents. Would it be fair to say that these are important for your model? Can you explain (or at least speculate), how and why? Any background for these choices would be very helpful.

Similarly, where you use different models for different neurons ("only non-zero during learnig" etc.), it would be nice to get an idea for your rationale. Why was it helpful to make different neurons different?

Page 12

"Plasticity": my general feedback for this statement is similar to that for membrane potential dynamics: can you please use several sentences to hint why you settled on this level of complexity for the dW/dt equation, and how you have arrived at your constant? Not in deep detail, but conceptually? What were your criteria? Say, for each neuron you use 5 different proxies for its activity: V, s, u, v, and x. Why do you need all five? Why 3 (V, s, filtered_s) would not be enough? Similarly, why u and v use two just slightly different taus? Is it to create an asymmetric waveform over time, or is it just a result of manual tweaking? I realize that it would be counter-productive to "justify" any of these micro-choices, but it would be very useful to have some basic understanding of what you were trying to achieve, and how you figured whether you achieved it.

"Heaviside function" - please remind the reader the definition of this function, in brackets

Reviewer #2: The manuscript by Maes, Barahona, and Clopath proposes a computational model that learns to produce spatiotemporal signals using a recurrent network model with Hebbian plasticity rules. Authors show that the model learns high-dimensional read-out spiking sequences on a behavioral time scale, with biologically plausible voltage-based STDP rules, and then claim that the architecture separates time and space into two different parts and this allows learning to bind space to time.

I believe this type of model approach is quite important and helpful to provide a better understanding of underlying principles in experimental observations. While interesting and potentially relevant for the study of the gap between behavioral and neural timescales, a substantial portion of the model designs and simulation analyses need to be presented with much better explanation of “meaning”, to clarify the rationale behind the model choice and the goal of each test. A major issue is that it is not clear what exactly can be learned from these results, what kind of problem can be solved with the current model (but cannot be solved by previous ones), and to which experimental observations each simulation result is relevant. I believe that some major issues, including the above, need to be addressed before publication can be recommended.

Major issues

1. Authors need to discuss specifically what is the difference between the current model and the previous ones. For example, learning of spatio-temporal sequence with RNN has already been implemented in previous studies. For example, what is the difference and advantage of the current model compared with the model in Chenkov, Sprekeler & Kempter, PloS Comput Biol 2017?

If authors think that the current model uses more biologically plausible components, as mentioned in abstract “… current computational models do not typically use realistic biologically plausible learning.”, there are some model studies that use realistic, biologically plausible learning rules. For example, see Park, Choi & Paik, Scientific Reports (2017) and also Lee et al. bioRxiv 525220 (2019).

In addition, there are a number of successful models for the spike sequence learning (e.g. tempotron by Sompolinsky). Thus it will be helpful that authors discuss these previous approaches and compare them with the current model, to clarify what kind of novelty the current model has.

2. How can you control the timescale of temporal backbone in the RNN? Or is there any critical factor that can control it? It needs to be explained how this timescale changes in the model and how it is relevant to real biological mechanism to handle different behavioral timescales across tasks. In the current model, conversion from the neural timescale (ms) to behavioral timescale (hundred ms) seems to strongly depend on the transmission delay. How and how much the timescale can vary in the model? Can you explain how to make much longer backbones than the current ones? Also what happens if the supervisor signal length varies?

3. In the intro, authors mention that “… the recurrent network encodes time and the read-out neurons encode ’space’. ’Space’ can mean spatial position, but it can also mean frequency or a more abstract state space”. I think the word “space” here is confusing, or not the best choice. Authors mentioned (in discussion) that “a set of read-out neurons that encode multiple dimensions of a signal, which we also call space”. Does it mean the spatial structure of feedforward connectivity, or structure of input pattern (whatever the contents are..)? I understand that authors intended to express a generalized “spatio-temporal” information, but this needs to be better described to avoid confusion.

4. How much information can be learned in the current network? i.e. what is the capacity of the current model network? Is it possible to train more than one non-Markov pattern in the same network? Is it biologically plausible, if your model uses a number of neurons only to learn one or just a couple of patterns?

5. What are the roles of inhibitory neurons and inhibitory plasticity? Is there any particular condition required for them, except for preventing runaways dynamics? A more detailed role of inhibitory population might be important in the model.

6. Can you try any test to show the robustness of the model? For example, how many synapses can be destroyed without losing the trained information? I guess destroying RNN-readout connections might be critical. Then, is there any evidence that this system is relatively robust than others?

**Have all data underlying the figures and results presented in the manuscript been provided?**

Reviewer #1: No: The authors promise to provide it after acceptance?

Reviewer #2: Yes

PLOS authors have the option to publish the peer review history of their article (what does this mean?). If published, this will include your full peer review and any attached files.

Reviewer #1: Yes: Arseny S. Khakhalin

Reviewer #2: No

---

## [Decision Letter · Decision Letter 1]

3 Dec 2019

Dear Dr Clopath,

Thank you very much for submitting your manuscript, 'Learning spatiotemporal signals using a recurrent spiking network that discretizes time', to PLOS Computational Biology. As with all papers submitted to the journal, yours was fully evaluated by the PLOS Computational Biology editorial team, and in this case, by independent peer reviewers. The reviewers appreciated the attention to an important topic but identified some aspects of the manuscript that should be improved. 

Specifically, both the reviewers were happy with the larger changes you made, but have some remaining smaller questions/concerns. These remaining issues can likely be addressed via discussion in the text rather than new data/results.

We would therefore like to ask you to modify the manuscript according to the review recommendations before we can consider your manuscript for acceptance. Your revisions should address the specific points made by each reviewer and we encourage you to respond to particular issues. 

Please note while forming your response, if your article is accepted, you may have the opportunity to make the peer review history publicly available. The record will include editor decision letters (with reviews) and your responses to reviewer comments. If eligible, we will contact you to opt in or out.raised.

- Supporting Information uploaded as separate files, titled 'Dataset', 'Figure', 'Table', 'Text', 'Protocol', 'Audio', or 'Video'.

We hope to receive your revised manuscript within the next 30 days. If you anticipate any delay in its return, we ask that you let us know the expected resubmission date by email at ploscompbiol@plos.org.

Sincerely,

Blake A. Richards

Associate Editor

PLOS Computational Biology

Kim Blackwell

Deputy Editor

PLOS Computational Biology

[LINK]

Reviewer's Responses to Questions

**Comments to the Authors:**

Reviewer #1: Overall, I am very happy with the edits, and I think the paper has become so much better with these additional analyses, and constructive rewrites! It is a pleasure to read; it provides a decent amount of background information, and answers most of the questions the readers might have.

My only concern is about the following two statements:

p 123: We set the initial values of the non-zero weights in the recurrent network such that the dynamics is irregular and asynchronous (i.e., a balanced network, see Methods for details).

p 408: The weights are initialized such that the network is balanced, i.e. it exhibits irregular asynchronous spiking dynamics.

As you can see, in the results you refer the reader to the Methods, but in the Methods, you essentially reiterate the same statement, and leave it at that. You don't describe why this network would be called "balanced" (I assume you could use a reference for that, as probably somebody used this word for chaotic before), you don't describe the criteria for "irregularity" (did you just observe it by eye, or have you automated it?), and you don't describe the actual process of "ensuring that it is balanced" (I'm guessing you just generated a network, tested it, and if it was bad, generated a different one - but it would be nice to state it). I would recommend that you add a sentence or two about that in the methods section (immediately following line 408).

Other than that, I'm happy with the paper, and recommend it for publication. Nice work!!

- Arseny

Reviewer #2: The authors have made a comprehensive and detailed revisions and addressed most of issues I raised. The revised manuscript is improved in some respects, particularly for the description of the robustness of the model and the modulation of the period of sequential dynamics. I have several questions and suggestions left, related to the capacity and of the model and its biological plausibility.

1. The authors show that the network can learn multiple patterns (Fig. 4). However, this model seems to require a number of readouts, which I don’t believe the optimal design of memory. For example, to memorize N patterns of the length L, how many readout neurons do you need? Can your model share the same readouts for pattern A and B? If not, is this biologically plausible model?

2. Because the network uses a voltage-based STDP, the pattern need to be trained multiple times in general (i.e. There is no one-shot learning mechanism in this model). Can you implement, or discuss any possible mechanism that address this issue? For example, synchronization between the temporal backbone and supervisor signal can make any changes of learning rate or efficiency?

3. In the current model, the RNN module that encodes time information and the readout module for spatial information are separated. Do you have a good explanation that this is biologically realistic design, or can you suggest any possible experiment for finding evidences of this structure in biological brain? Then, what do you think is the advantage of separated circuits for spatial and temporal information? What happens when the network learns a spatio-temporally correlated information?

4. Still I find the current model is very similar with previous models in some sense, except for some biological components. Can you better clarify the need of current model, with examples of tasks that cannot be performed with previous designs?

**Have all data underlying the figures and results presented in the manuscript been provided?**

Reviewer #1: Yes

Reviewer #2: Yes

PLOS authors have the option to publish the peer review history of their article (what does this mean?). If published, this will include your full peer review and any attached files.

Reviewer #1: Yes: Arseny S. Khakhalin

Reviewer #2: No

---

## [Editor Report · Decision Letter 2]

13 Dec 2019

Dear Dr Clopath,

We are pleased to inform you that your manuscript 'Learning spatiotemporal signals using a recurrent spiking network that discretizes time' has been provisionally accepted for publication in PLOS Computational Biology.

In the meantime, please log into Editorial Manager at https://www.editorialmanager.com/pcompbiol/, click the "Update My Information" link at the top of the page, and update your user information to ensure an efficient production and billing process.

One of the goals of PLOS is to make science accessible to educators and the public. PLOS staff issue occasional press releases and make early versions of PLOS Computational Biology articles available to science writers and journalists. PLOS staff also collaborate with Communication and Public Information Offices and would be happy to work with the relevant people at your institution or funding agency. If your institution or funding agency is interested in promoting your findings, please ask them to coordinate their releases with PLOS (contact ploscompbiol@plos.org).

Thank you again for supporting Open Access publishing. We look forward to publishing your paper in PLOS Computational Biology.

Sincerely,

Blake A. Richards

Associate Editor

PLOS Computational Biology

Kim Blackwell

Deputy Editor

PLOS Computational Biology

---

## [Editor Report · Acceptance letter]

10 Jan 2020

PCOMPBIOL-D-19-01145R2 

Learning spatiotemporal signals using a recurrent spiking network that discretizes time

Dear Dr Clopath,

I am pleased to inform you that your manuscript has been formally accepted for publication in PLOS Computational Biology. Your manuscript is now with our production department and you will be notified of the publication date in due course.

With kind regards,

Sarah Hammond
